# A neutrophil–B-cell axis impacts tissue damage control in a mouse model of intraabdominal bacterial infection via Cxcr4

**Riem Gawish**[1,2], **Barbara Maier**[1,2], **Georg Obermayer**[2,3], **Martin L Watzenboeck**[1,2], **Anna-Dorothea Gorki**[1,2], **Federica Quattrone**[1,2], **Asma Farhat**[1,2], **Karin Lakovits**[1], **Anastasiya Hladik**[1], **Ana Korosec**[1], **Arman Alimohammadi**[4], **Ildiko Mesteri**[5†], **Felicitas Oberndorfer**[5], **Fiona Oakley**[6], **John Brain**[6], **Louis Boon**[7], **Irene Lang**[4], **Christoph J Binder**[2,3], **Sylvia Knapp**[1,2]*

[1]Department of Medicine I, Laboratory of Infection Biology, Medical University Vienna, Vienna, Austria; [2]Ce-M-M-, Research Center for Molecular Medicine of the Austrian Academy of Sciences, Vienna, Austria; [3]Department of Laboratory Medicine, Medical University of Vienna, Vienna, Austria; [4]Department of Medicine II, Division of Cardiology, Medical University of Vienna, Vienna, Austria; [5]Department of Pathology, Medical University Vienna, Vienna, Austria; [6]Newcastle Fibrosis Research Group, Biosciences Institute, Newcastle University, Newcastle, United Kingdom; [7]Polypharma Biologics, Utrecht, Netherlands

*For correspondence: Sylvia.knapp@meduniwien.ac.at

Present address: †Pathology Überlingen, Überlingen, Germany

**Abstract** Sepsis is a life-threatening condition characterized by uncontrolled systemic inflammation and coagulation, leading to multiorgan failure. Therapeutic options to prevent sepsis-associated immunopathology remain scarce. Here, we established a mouse model of long-lasting disease tolerance during severe sepsis, manifested by diminished immunothrombosis and organ damage in spite of a high pathogen burden. We found that both neutrophils and B cells emerged as key regulators of tissue integrity. Enduring changes in the transcriptional profile of neutrophils include upregulated Cxcr4 expression in protected, tolerant hosts. Neutrophil Cxcr4 upregulation required the presence of B cells, suggesting that B cells promoted disease tolerance by improving tissue damage control via the suppression of neutrophils' tissue-damaging properties. Finally, therapeutic administration of a Cxcr4 agonist successfully promoted tissue damage control and prevented liver damage during sepsis. Our findings highlight the importance of a critical B-cell/neutrophil interaction during sepsis and establish neutrophil Cxcr4 activation as a potential means to promote disease tolerance during sepsis.

## Editor's evaluation

In this elegant animal experiment on a novel mouse model, the authors' aim was to investigate the mechanisms of long-lasting disease tolerance to identify potential targets to prevent organ damage in sepsis. The results showed that interaction between B cells and neutrophils plays a pivotal role in preventing tissue damage and modulation, rather than upregulation of the Cxcr4 expression of neutrophils requires the presence of B cells, and this promotes disease tolerance by improving tissue damage control via the suppression of neutrophils' tissue damaging properties. Furthermore, treatment with a Cxcr4-agonist successfully replicated the tissue tolerance phenotype and prevented organ damage. These results could have important clinical and research implications by unveiling

further Cxcr-dependent mechanisms and potentially can lead to the development of a novel therapeutic approach to sepsis.

## Introduction

Sepsis is a life-threatening condition triggered by severe infections with bacteria, viruses, or fungi. In spite of the successful use of antimicrobial therapies, mortality rates remain high, with up to 50% (*Mayr et al., 2014*; *Holzheimer et al., 1991*). The main determinant of sepsis-associated mortality is rarely the pathogen, but instead the combination of dysregulated systemic inflammation, immune paralysis, and hemostatic abnormalities that together cause multiorgan failure (*Singer et al., 2016*). Upon pathogen sensing, ensuing inflammation promotes the activation of coagulation, which in turn generates factors that further amplify inflammation, thus creating a vicious, self-amplifying cycle. These events result in systemic inflammation and the widespread formation of microvascular thrombi, that together cause vascular leak, occlusion of small vessels, and eventually multiorgan failure (*Semeraro et al., 2015*; *Gando et al., 2016*). At the same time, sepsis goes along with a state of immune cell dysfunction characterized by immune cell exhaustion, enhanced apoptosis and impaired antigen presentation, cytokine production, and pathogen killing (*Hotchkiss et al., 2013*). Whether a patient suffering from sepsis enters this fatal circuit of immunopathology or instead is able to maintain vital organ functions and survives sepsis is not well understood (*Angus and van der Poll, 2013*; *Levy et al., 2003*; *Kotas and Medzhitov, 2015*).

The potent immunogen lipopolysaccharide (LPS) is an important driver of inflammation during Gram-negative sepsis, due to its ability to activate toll-like receptor (TLR)–4 (*Beutler, 2000*; *Hoshino et al., 1999*) and caspase-11 (*Hagar et al., 2013*; *Kayagaki et al., 2013*). The state of sepsis-induced immune paralysis can be partially recapitulated in vitro by the phenomenon of LPS-tolerance, which is characterized by an unresponsive state of myeloid cells after repeated TLR stimulation (*de Vos et al., 2009*; *Fan and Cook, 2004*). In contrast, the concept of 'disease tolerance' describes a poorly studied, yet essential host defense strategy, on top of the well understood strategies of avoidance and resistance (*Medzhitov et al., 2012*). While avoidance means preventing pathogen exposure and infection, and resistance aims to more efficiently reduce the pathogen load in the course of an established infection, disease tolerance involves mechanisms, which minimize the detrimental impact of infection irrespective of the pathogen burden, thus improving host fitness despite the infection for example by improving tissue damage control (*Medzhitov et al., 2012*; *Martins et al., 2019*). Importantly, LPS-tolerance can impact both, pathogen resistance and/or disease tolerance. To this end, a number of mechanisms that shape the process of disease tolerance have been suggested, including alterations in cellular metabolism, DNA damage response, tissue remodeling or oxidative stress (*Martins et al., 2019*). However, little is known about the specific contribution of immune cells to disease tolerance during severe infections, and therapeutic options to increase disease tolerance are limited due to a lack of knowledge about detailed molecular and cellular tolerance mechanisms (*Angus and van der Poll, 2013*; *Levy et al., 2003*; *Kotas and Medzhitov, 2015*).

In this study, we investigated mechanisms of disease tolerance and tissue damage control by comparing tolerant and sensitive hosts during a severe bacterial infection. While sensitive animals developed severe coagulopathy and tissue damage during sepsis, tolerant animals showed improved tissue damage control and were able to maintain tissue integrity in spite of a high bacterial load. Disease tolerance was induced by the prior exposure of animals to a single, low-dose of LPS and could be uncoupled from LPS-induced suppression of cytokine responses. We provide evidence for a deleterious and organ-damaging interaction between B cells and neutrophils during sepsis in sensitive animals, while in tolerant animals, neutrophils and B cells jointly orchestrated tissue protection during sepsis, which was associated with transcriptional reprogramming of neutrophils and B cell dependent upregulation of neutrophil Cxcr4. Our data suggest that B cells can modulate the tissue damaging properties of neutrophils by influencing neutrophil Cxcr4 signaling. Consequently, the administration of a Cxcr4 agonist prevented sepsis-associated microthrombosis and resulting tissue damage, thereby exposing a potential therapeutic strategy to foster tissue damage control in severe sepsis.

## Results

### LPS pre-exposure induces long-term disease tolerance during Gram-negative sepsis

LPS pre-exposure has been shown to prevent hyperinflammation during fatal endotoxemia via the induction of LPS-tolerance (*López-Collazo and del Fresno, 2013*) and to improve the outcome of cecal ligation and puncture (CLP)-induced sepsis by promoting bacterial clearance (*Márquez-Velasco et al., 2007*). To investigate mechanisms of tissue damage control and disease tolerance during bacterial sepsis, we aimed for a setup with a stable pathogen load which would allow us to only interfere with sepsis-associated organ failure. We thus challenged mice intravenously (i.v.) with a subclinical dose of LPS 1-day, 2 weeks, 5 weeks, or 8 weeks, respectively, prior to the induction of Gram-negative sepsis by intraperitoneal injection of the virulent *E. coli* strain O18:K1. While LPS pretreatment 24 hr prior

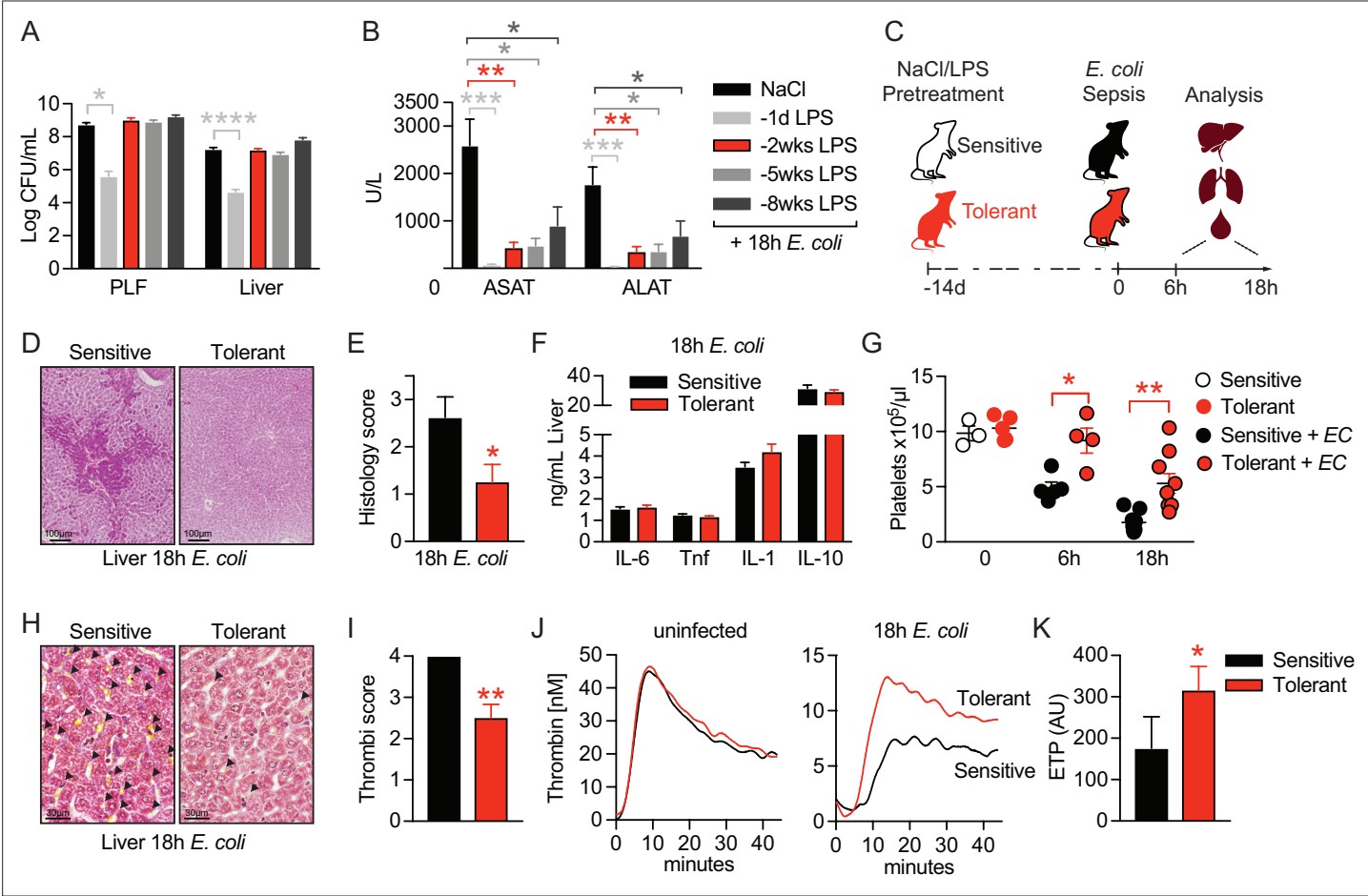

**Figure 1.** LPS pre-exposure induces long-term disease tolerance during Gram-negative sepsis. (**A**) *E. coli* colony forming units (CFU) 18 hr post-infection (p.i.) in peritoneal lavage fluid (PLF) and liver of mice, which were pretreated with lipopolysaccharide (LPS) or saline (NaCl) at depicted time points before infection. (**B**) Aspartate aminotransferase (ASAT) and alanine aminotransferase (ALAT) plasma levels at 18 hr p.i. of mice, pretreated with LPS or NaCl at indicated time points. (**C**) Schematic depiction of the treatment procedure and endpoints. (**D–E**) H&E staining and pathology scores of liver sections from mice pretreated with LPS or NaCl 2 weeks earlier, and infected with *E. coli* for 18 hr. (**F**) Liver cytokine levels at 18 hr p.i. (**G**) Blood platelet counts at 2 weeks post LPS/NaCl pretreatment, and 6 hr or 18 hr p.i. (**H–I**) Martius, scarlet and blue (MSB) fibrin staining of liver sections and scoring of liver microthrombi at 18 hr p.i. (**J**) In vitro thrombin generation capacity of plasma from LPS/NaCl pretreated uninfected and infected mice. (**K**) Endogenous thrombin potential (ETP) of plasma samples 18 hr p.i. Data in (**A**) and (**B**) are pooled from 2 to 3 independent experiments (n=6–7/experimental group). Data in (**G**) are pooled from two independent experiments (n=1–3/experimental group for uninfected and 4–5 for infected mice). All other data are representative for two or more independent experiments (n=8/experimental group). All data are and presented as mean +/-SEM. * p≤0.05, ** p≤0.01, *** p≤0.001 and **** p≤0.0001.

The online version of this article includes the following figure supplement(s) for figure 1:

**Figure supplement 1.** LPS pre-exposure induces long-term disease tolerance during Gram-negative sepsis.

to infection significantly improved pathogen clearance, any longer period (i.e. 2–8 weeks) between LPS administration and infection did not affect the bacterial load when compared to control mice (*Figure 1A*, *Figure 1—figure supplement 1A*). Importantly, though all LPS pretreated groups were substantially protected from sepsis-associated tissue damage, illustrated by the absence of elevated liver transaminase (aspartate aminotransferase [ASAT] and alanine aminotransferase [ALAT]) plasma levels (*Figure 1B*). Thus, short-term (24 hr) LPS pre-exposure improved resistance to infection and consequently tissue integrity, while long-term (2–8 weeks) LPS pre-exposure enabled the maintenance of tissue integrity irrespective of a high bacterial load, which per definition resembles disease tolerance.

To dissect the underlying mechanism of tissue damage control in disease tolerant mice, we thus performed all subsequent experiments by treating mice with either LPS or saline two weeks prior to bacterial infection, allowing us to compare tolerant with sensitive hosts. Mice were either sacrificed two weeks after LPS pretreatment to assess changes in tolerant hosts prior to infection, or six to 18 hr after *E. coli* infection to determine early (6 hr) or late inflammation and organ damage (18 hr), respectively, during sepsis (*Figure 1C*). Doing so, we observed that organ protection (*Figure 1B*) was associated with the absence of liver necrosis (*Figure 1D and E*), while inflammatory cytokine and chemokine levels were indistinguishable between sensitive and tolerant mice 18 hr post-infection (p.i.) (*Figure 1F*, *Figure 1—figure supplement 1B and C*). A major cause of organ damage during sepsis is the disseminated activation of coagulation, which is characterized by systemic deposition of micro-thrombi and substantial platelet consumption, resulting in a critical reduction in tissue perfusion (*Semeraro et al., 2015*; *Gando et al., 2016*; *Angus and van der Poll, 2013*). While we discovered a severe decline in platelet numbers upon *E. coli* infection in sensitive mice, tolerant mice maintained significantly higher blood platelet counts (*Figure 1G*) and, in sharp contrast to sensitive animals, showed almost no deposition of micro-thrombi in liver (*Figure 1H and I*) and lung sections (*Figure 1—figure supplement 1D*), indicating that tissue damage control occurred systemic and not organ specific. Considering that LPS exposure itself can impact coagulation factor levels and blood platelet numbers (*Asakura et al., 2003*; *Ohtaki et al., 2002*), we importantly found similar platelet counts in sensitive and tolerant mice at the onset of *E. coli* infection (2 weeks post-LPS) (*Figure 1G*). In addition, we did not detect any indication for an altered coagulation potential in tolerant mice before sepsis induction, as both groups showed a similar plasma thrombin generation potential prior to infection (*Figure 1J* left panel, *Figure 1—figure supplement 1E*). However, compared to sensitive animals, the thrombin generation capacity was only preserved in tolerant mice after infection (18 hr p.i.), suggesting that tolerance mechanisms prevented sepsis-associated consumption coagulopathy (*Figure 1J* right panel and 1 K). Taken together, low-dose LPS pretreatment prevented the formation of micro-thrombi and induced a long-lasting state of disease tolerance during subsequent sepsis.

## B cells control tissue damage during sepsis independent of early inflammatory responses

Considering the long-term protective effect of LPS pre-exposure in tolerant animals, we next tested the possibility that long-lived immune cells like lymphocytes might impact tissue damage control during sepsis. Strikingly, the absence of lymphocytes, as in *Rag2* deficient (*Rag2*-/-) animals, already resulted in profoundly reduced liver damage upon bacterial infection of naïve, sensitive mice and fully abrogated further LPS-induced tissue protection without affecting the bacterial load (*Figure 2A*, *Figure 2—figure supplement 1A and B*). This indicated that lymphocytes on the one hand importantly contributed to the sensitivity of animals to sepsis-associated organ damage in naïve mice, and on the other hand were essential in mediating LPS-induced tissue protection in tolerant hosts. In support of this, adoptive transfer of splenocytes into *Rag2*-/- mice re-established LPS-induced tissue damage control (*Figure 2A*). Depleting CD8 or CD4 T cells, respectively, prior to LPS exposure (*Figure 2B* and *Figure 2—figure supplement 1C*) neither affected the difference between sensitive and tolerant animals to organ damage (*Figure 2C*) nor the bacterial load (*Figure 2—figure supplement 1D*) upon *E. coli* infection. In contrast, B cell deficiency (J_HT mice) fully prevented the development of tissue damage during sepsis irrespective of tolerance induction (*Figure 2D*) and despite a high bacterial load (*Figure 2—figure supplement 1E*). Furthermore, adoptive transfer of B cells into *Rag2*-/- mice (*Figure 2E*) aggravated liver damage upon *E. coli* infection in sensitive and reestablished tissue protection in tolerant mice (*Figure 2F*). These findings indicated that B cells, but not T cells, played an

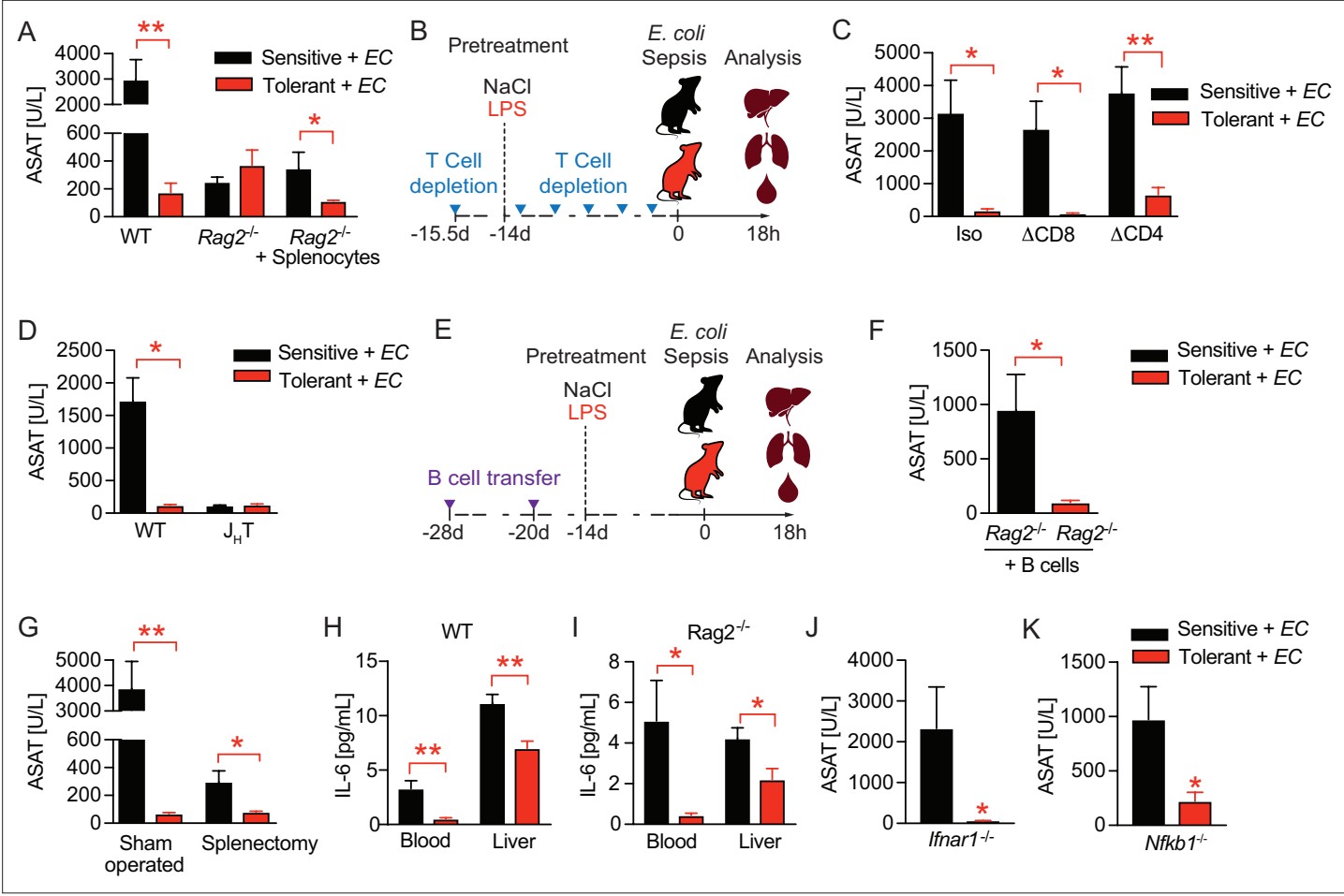

**Figure 2.** B cells regulate disease tolerance during sepsis independent of early inflammatory responses. (**A**) Aspartate aminotransferase (ASAT) plasma levels at 18 hr p.i. with *E. coli*, of lipopolysaccharide (LPS) or NaCl pretreated wildtype or lymphocyte deficient mice (*Rag2⁻/⁻*), which have received either phosphate-buffered saline (PBS) or splenocytes i.v. 3 weeks prior infection. (**B**) Schematic depiction of the treatment procedure for T cell depletion experiments. (**C**) ASAT plasma levels 18 hr p.i. with *E. coli* of mice, which were depleted from CD4⁺ or CD8⁺ T cells prior to LPS or NaCl pretreatment. (**D**) ASAT plasma levels of LPS or NaCl pretreated wildtype or B cell deficient (JₕT) mice at 18 hr p.i. with *E. coli*. (**E**) Schematic depiction of the treatment procedure for splenocyte and B cell transfer experiments. (**F**) ASAT plasma levels of LPS or NaCl pretreated *Rag2⁻/⁻* mice at 18 hr p.i. with *E. coli*, which have been reconstituted with bone marrow derived B cells 3 weeks before infection. (**G**) ASAT plasma levels at 18 hr p.i. with *E. coli* of LPS or NaCl pretreated mice, which were splenectomized or sham operated 1 week before LPS or NaCl pre-exposure (i.e. 3 weeks before infection). (**H–I**) IL-6 levels in plasma and liver of NaCl or LPS pretreated wildtype or *Rag2⁻/⁻* mice at 6 hr p.i. with *E. coli*. (**J**) ASAT plasma levels of NaCl or LPS pretreated *Ifnar1⁻/⁻* mice at 18 hr p.i. with *E. coli*. (**K**) ASAT plasma levels of NaCl or LPS pretreated *Nfkb1⁻/⁻* mice at 18 hr p.i. with *E. coli*. Data in (**A**) and (**G–J**) are representative out of 2–3 experiments (n=3–8/experimental group). Data in (**D**) and (**K**) are pooled from 2 independent experiments (n=2–7/experimental group). Data in (**C**) and (**F**) are from a single experiment (n=6–8/group). Data are and presented as mean +/-SEM. * p≤0.05 and ** p≤0.01.

The online version of this article includes the following figure supplement(s) for figure 2:

**Figure supplement 1.** B cells regulate disease tolerance during sepsis independent of early inflammatory responses.

ambiguous role as they were involved in both, sepsis-associated organ damage and the establishment of LPS-triggered disease tolerance. We then tested if splenectomy would replicate the protective effects of full B cell deficiency during sepsis and interestingly found that splenectomy was associated with reduced liver damage in naïve, sensitive mice, which is in line with other studies (*Agarwal et al., 1972*; *Karanfilian et al., 1983*), but, in contrast to complete lymphocyte deficiency, not sufficient to abrogate LPS-induced tissue protection in tolerant animals (*Figure 2G* and *Figure 2—figure supplement 1F*). This suggested that mature splenic B cells contributed to tissue damage during severe infections, while other, not spleen derived, B cell compartments were instrumental in driving disease tolerance.

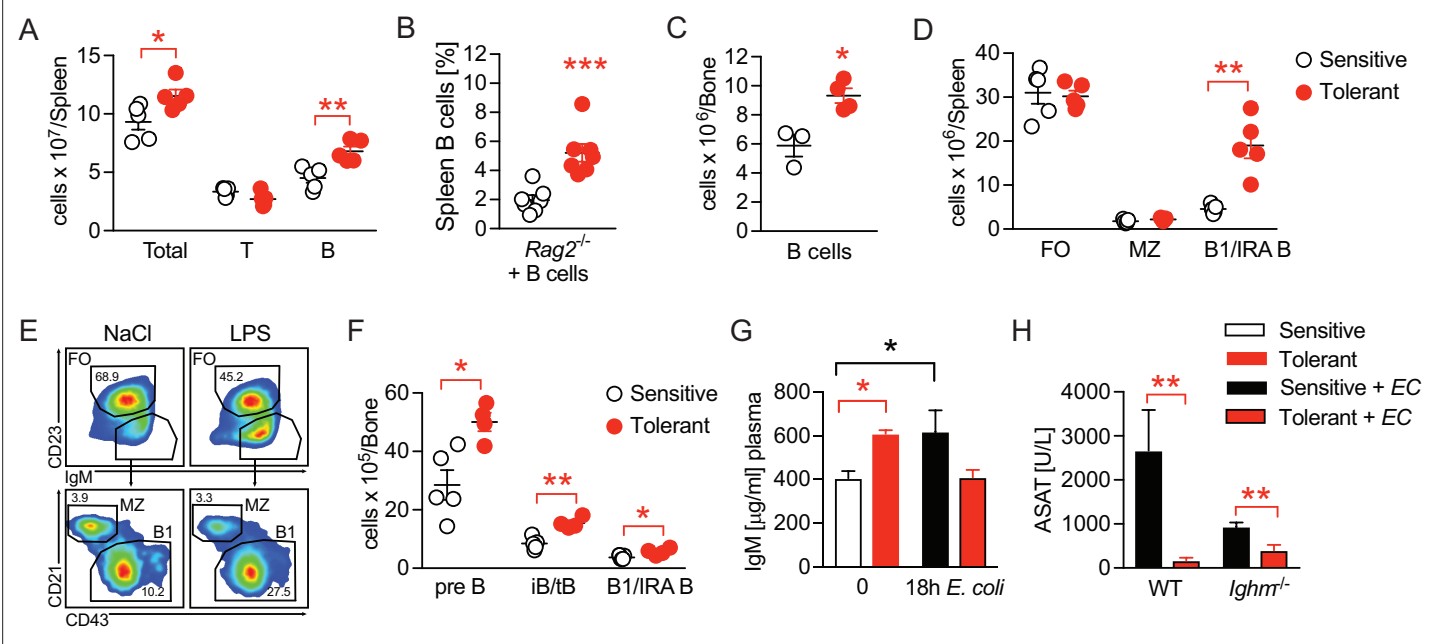

**Figure 3.** Disease tolerance is associated with rearranged B cell compartments. (**A**) Flow-cytometric analysis of B and T cells in the spleen of mice treated with lipopolysaccharide (LPS) or NaCl 2 weeks earlier. (**B**) Flow-cytometric analysis of B cells in spleens of *Rag2⁻/⁻* mice treated with LPS or NaCl 2 weeks earlier, and reconstituted with GFP⁺ B cells before LPS/NaCl. (**C**) CD19⁺ B cells per femur of mice treated with NaCl or LPS 2 weeks earlier. (**D**) Flow-cytometric analysis of FO, MZ and B1/IRA B cells in spleens of mice treated with LPS or NaCl 2 weeks earlier. (**E**) Gating strategy for splenic B cell subsets. (**F**) Flow-cytometric analysis of Pre-B, iB/tB and B1/IRA B cells in the bone marrow of mice treated with LPS or NaCl 2 weeks earlier. (**G**) IgM plasma levels in NaCl or LPS pretreated uninfected mice and 18 hr p.i. with *E. coli*. (**H**) Aspartate aminotransferase (ASAT) plasma levels of NaCl or LPS pretreated WT and *Ighm⁻/⁻* mice 18 hr p.i. with *E. coli*. Data in (**A**) and (**C–F**) and are representative out of 2–3 experiments (n=3–8/experimental group). Data in (**G–H**) are pooled from two independent experiments (n=3–7/experimental group). Data in (**B**) are from a single experiment (n=7/group) and all data are and presented as mean +/-SEM. * p≤0.05 and ** p≤0.01.

The online version of this article includes the following figure supplement(s) for figure 3:

**Figure supplement 1.** Disease tolerance is associated with rearranged B cell compartments.

Given that B cells were shown to promote early production of proinflammatory cytokines such as IL-6 during sepsis in a type I IFN dependent manner (*Kelly-Scumpia et al., 2011*), we next investigated if LPS pretreatment improved tissue damage control by dampening B cell driven inflammatory responses. Six hours post *E. coli* infection, we found tolerant wild type mice to exhibit lower IL-6 levels in blood and liver (*Figure 2H*), as well as lower amounts of important regulators of peritoneal leukocyte migration (*Rajarathnam et al., 2019*; *Bianconi et al., 2018*), like CXCL1 and CCL2 (*Figure 2—figure supplement 1G*) when compared to sensitive control mice, a phenotype which is reminiscent of LPS-tolerance. However, lymphocyte deficient *Rag2⁻/⁻* animals, in whom tissue damage control could not be improved by LPS preexposure (*Figure 2A* and *Figure 2—figure supplement 1A*), showed comparable reductions in these mediators of early inflammation in response to LPS pretreatment (*Figure 2I* and *Figure 2—figure supplement 1H*). Furthermore, tissue damage control and LPS-induced disease tolerance during sepsis was induced independent of interferon-α/β receptor (IFNAR) signaling (*Figure 2J* and *Figure 2—figure supplement 1I*) and the anti-inflammatory NF-κB subunit p50 (NF-κB1) (*Figure 2K* and *Figure 2—figure supplement 1J*), which has been shown to mediate the suppression of cytokine production during endotoxin tolerance in vitro (*Fan and Cook, 2004*; *Ziegler-Heitbrock, 2001*). These data suggested that in tolerant hosts, B cells contributed to tissue protection during sepsis, and that an LPS mediated modulation of early inflammation is unlikely to explain these protective effects.

## Disease tolerance is associated with rearranged B cell compartments

We next compared the B cell compartment in sensitive and tolerant mice and analyzed different B cell populations in spleen and bone marrow 2 weeks after saline or LPS treatment. Tolerance was

associated with a mild increase in spleen weight (*Figure 3—figure supplement 1A*) and total spleen cell counts, together with an expansion of B cell numbers (*Figure 3A*). Of note, tolerance-induction by LPS treatment even caused an expansion of transplanted B cells in spleens of *Rag2*$^{-/-}$ animals (*Figure 3B*). In parallel, while the total number of bone marrow cells remained indistinguishable (*Figure 3—figure supplement 1B*), we found bone marrow B cell numbers increased in tolerant mice, as compared to sensitive controls (*Figure 3B*). Furthermore, analysis of B cell subsets revealed an increase of B1 B cell numbers (IgM$^{hi}$ CD23$^-$ CD43$^{hi}$ CD21$^-$), a subset that also includes the B1-like innate response activator (IRA) B cells, in spleen and bone marrow (*Figure 3D–F* and *Figure 3—figure supplement 1C*), as well as elevated numbers of Pre-B and immature and transitional B cells (iB/tB) in the bone marrow (*Figure 3F* and *Figure 3—figure supplement 1C*). In sharp contrast, total numbers of follicular (FO) and marginal zone (MZ) B cells did not change upon tolerance induction (*Figure 3D*).

B1 B and IRA B cell-derived IgM was shown earlier to exert tissue protective properties and has been proposed as a possible mechanism of disease tolerance (*Márquez-Velasco et al., 2007*; *Rauch et al., 2012*; *Weber et al., 2014*; *Ha et al., 2006*; *Murakami et al., 1994*; *Baumgarth, 2016*; *Krautz et al., 2018*). In line with this, we found elevated plasma IgM levels in tolerant mice prior to infection, which–in contrast to sensitive, control animals–returned to baseline during sepsis, indicating LPS-induced induction of IgM, and consumption of IgM during sepsis in tolerant animals (*Figure 3G*). We therefore tested if IgM was an essential soluble mediator responsible for the tissue protection in tolerant mice. Unexpectedly though, mice lacking soluble IgM developed less severe organ damage, and LPS-pretreatment still induced tissue protection during sepsis (*Figure 3H*, *Figure 3—figure supplement 1D*). Taken together, tissue damage control was associated with long-term changes in the B cell compartments in the spleen and bone marrow, and the B cells' tissue protective effects in tolerant mice occurred in an IgM-independent manner.

## B cells impact neutrophils, the key effectors driving sepsis-induced tissue damage

Having determined the importance of B cells in mediating tissue damage control during sepsis, and having ruled out B-cell mediated inflammation or IgM effects as driving forces, we wondered if B cells might impact on the functionalities of other immune effector cells in sepsis. To assess the tissue damaging potential of candidate effector cells in our model, we depleted neutrophils (ΔPMN) (*Figure 4—figure supplement 1A*), platelets (ΔPlt) (*Figure 4—figure supplement 1B*), or monocytes and macrophages (ΔM) (*Figure 4—figure supplement 1C*) in sensitive and tolerant mice, respectively, prior to *E. coli* infection. Surprisingly, monocyte and macrophage depletion neither influenced sepsis-induced tissue damage in sensitive animals nor did it impact on LPS-induced tissue damage control (*Figure 4A*), suggesting that classical LPS-tolerance is not the sole reason for protection. Platelet or neutrophil depletion, in contrast, already exerted tissue protective effects in both groups, illustrated by greatly reduced ASAT levels in sensitive and tolerant mice (*Figure 4A*). However, while LPS-pretreatment still enhanced tissue protection in ΔPlt mice, it did not result in any additive beneficial effects in ΔPMN animals (*Figure 4A*), similar to what we had observed upon B cell deficiency (*Figure 2D*). These data support the reported role of platelets and neutrophils in promoting tissue damage during sepsis (*Semeraro et al., 2015*; *Gando et al., 2016*) and proved neutrophils to be key effector cells of tissue protection in tolerant animals. Of note, no significant impact on the pathogen load was detectable in any of the groups (*Figure 4—figure supplement 1D*).

As the full deficiency of either B cells or neutrophils abrogated organ damage during *E. coli* sepsis and LPS-induced protection could be re-established by adoptive transfer of B cells into *Rag2*$^{-/-}$ mice, we hypothesized an alliance between neutrophils and B cells in tissue damage control during sepsis. In steady state, up to 70% of CD45$^+$ bone marrow cells are composed of B cells and neutrophils, where both populations constitutively reside and mature by sharing the same niche (*Yang et al., 2013*). We therefore first analyzed bone marrow B cell and neutrophil dynamics after LPS challenge, and discovered substantial stress-induced granulopoiesis, peaking around day four post LPS exposure, while B cells were regulated in a reciprocal fashion as they vanished by day 4 post LPS injection, to then increase and remain elevated two weeks post LPS treatment (*Figures 4B and 3C*) in tolerant, as compared to sensitive animals. At the same time total and relative neutrophil numbers in the bone marrow remained slightly reduced in tolerant wild type mice, but elevated in the absence of B cells (*Figure 4C* and *Figure 4—figure supplement 1E*).

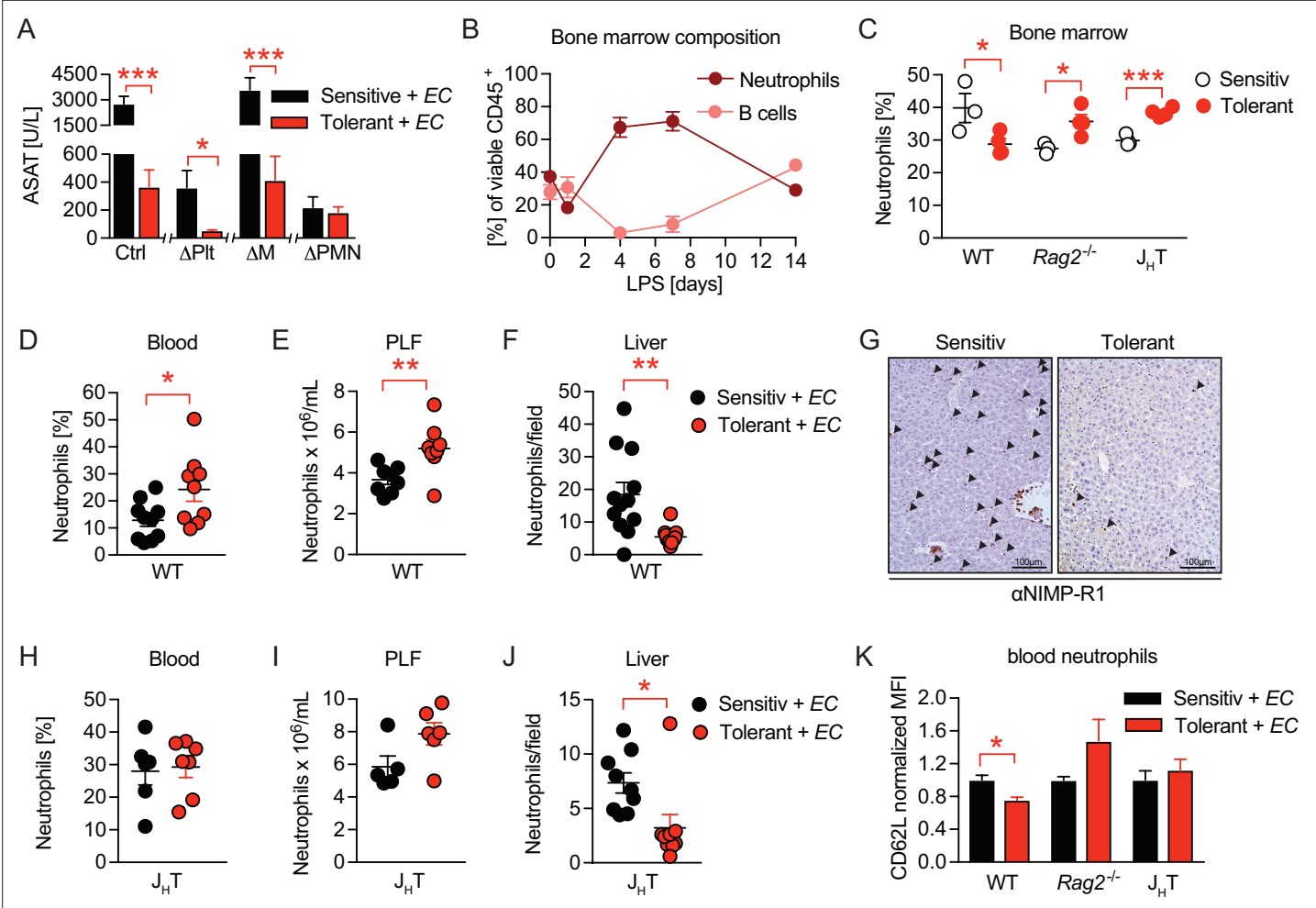

**Figure 4.** B cells impact neutrophils, the key effectors driving sepsis-induced tissue damage. (**A**) Aspartate aminotransferase (ASAT) plasma levels 18 hr p.i. with *E. coli* in lipopolysaccharide (LPS) or NaCl pretreated mice, in which platelets, monocytes/macrophages, or neutrophils, respectively, were depleted before infection. (**B**) Flow-cytometric analysis of bone marrow neutrophils and B cells after i.v. administration of LPS at time = 0 hr. (**C**) Flow-cytometric analysis of neutrophils in the bone marrow of wildtype, *Rag2^-/-^* and J$_H$T mice 2 weeks after LPS or NaCl treatment. (**D–E**) Flow-cytometric analysis of neutrophils of wildtype mice pre-treated with NaCl or LPS, respectively, and infected for 18 hr with *E. coli*, in blood (**D**) and peritoneal lavage fluid (PLF) (**E**). (**F–G**) Quantification of (**F**) immunohistological staining for NIMP-R1$^+$ cells on liver sections (**G**) of mice pretreated with NaCl or LPS, respectively, and infected with *E. coli* for 18 hr. (**H–J**) Flow-cytometric analysis of neutrophils 18 hr p.i. with *E. coli* in blood (**H**), PLF (**I**), and liver (**J**) of J$_H$T mice. (**K**) Flow-cytometric analysis of blood neutrophil CD62L expression of WT, *Rag2^-/-^* and J$_H$T mice at 18 hr p.i. with *E. coli*. Data in (**A**) shown for the control group and neutrophil depletion are pooled from two independent experiments (n=4–6/experimental group), platelet and monocyte/macrophage depletion represent a single experiment (n=8/group). Data in (**B**), (**D–E**), (**F**), and (**J**) are pooled from two independent experiments (n=4–8/experimental group). Data in (**H–I**) are representative of two experiments (n=5–8/group). Data in (**K**) are from a single experiment (n=4–8/group). All data are presented as mean +/-SEM. * p≤0.05, ** p≤0.01 and *** p≤0.001.

The online version of this article includes the following figure supplement(s) for figure 4:

**Figure supplement 1.** B cells impact neutrophils, the key effectors driving sepsis-induced tissue damage.

Next, we assessed differences between sensitive and tolerant animals in infection-induced peripheral neutrophil migration and abundance, depending on the presence or absence of lymphocytes or B cells, respectively. While tissue protection was associated with elevated neutrophil abundance in blood and peritoneal lavage fluid (PLF) of septic wild type mice (*Figure 4D and E*), neutrophil extravasation into tissues such as the liver and lung were substantially reduced (*Figure 4F–G* and *Figure 4—figure supplement 1F*). In both *Rag2^-/-^* and J$_H$T mice, LPS pretreatment did not cause increased blood neutrophils nor a significant accumulation in the PLF during sepsis (*Figure 4H–I* and *Figure 4—figure supplement 1G and H*). However, infection-induced neutrophil migration into liver tissue was still reduced after LPS pretreatment in J$_H$T mice (*Figure 4J*), but not in *Rag2^-/-^* animals (*Figure 4—figure*

*supplement 1I*). This suggested that in tolerant animals, B cells might affect systemic neutrophil trafficking and turnover after LPS-pre-exposure, whereas the suppressed neutrophil extravasation to the livers of tolerant mice occurred independent of B cells. In support of this idea, we discovered that blood neutrophils of tolerant mice expressed lower CD62L levels upon infection than those of sensitive controls, and that this phenotype required the presence of B cells (*Figure 4K*). While CD62L has been studied extensively for its importance in neutrophil adhesion and rolling over the vascular endothelium (*Ivetic, 2018*), a recent study has identified decreased CD62L expression indicative of neutrophil aging, a process that is counteracted by Cxcr4 signaling, the master regulator of neutrophil trafficking between the bone marrow and the periphery (*Adrover et al., 2019*; *Eash et al., 2010*; *Martin et al., 2003*). Based on these findings, we hypothesized that B cells might regulate sensitivity and tolerance during sepsis by affecting the functional plasticity and tissue damaging properties of neutrophils.

## Neutrophil tissue damaging properties are modulated by bone marrow B cells via Cxcr4

To assess the functional alterations in neutrophils, which confer tissue protection and tolerance during sepsis, we sorted neutrophils from the blood and bone marrow of sensitive and tolerant mice, i.e., 2 weeks post NaCl or LPS treatment but prior to infection, and performed RNA sequencing (*Figure 5—figure supplement 1A*). Despite the supposedly short life span of neutrophils, tolerant mice exhibited a sustained transcriptional reprogramming of the neutrophil pool. Principal component analysis of the 1000 most variable genes revealed clustering of neutrophils according to the site of sampling (*Figure 5A*), likely reflecting the heterogeneity of neutrophils in bone marrow versus mature cells in blood (*Semerad et al., 2002*; *Evrard et al., 2018*). Samples further separated according to treatment (*Figure 5A*) and we identified a substantial number of tolerance associated, differentially expressed genes (DEGs) in both, bone marrow and blood neutrophils (*Figure 5—figure supplement 1B, C and D*; *Supplementary files 1 and 2*). Gene ontology (GO) enrichment analysis of blood neutrophil DEGs exhibited an enrichment of genes associated with immunity and defense responses (*Figure 5B*). Strikingly, bone marrow neutrophils of tolerant mice showed an enrichment of genes associated with cell migration, trafficking, and chemotaxis (*Figure 5C*), such as genes involved in Cxcl12/Cxcr4 signaling including *Cxcr4* itself (*Figure 5—figure supplement 1E*).

Considering the reported importance of Cxcr4 signaling in neutrophil retention in the bone marrow and their release to the periphery (*Adrover et al., 2019*; *Eash et al., 2010*; *Martin et al., 2003*), we verified an upregulation of Cxcr4 on bone marrow derived neutrophils of tolerant mice compared to sensitive control mice on a transcriptional (*Figure 5D*) and protein level (*Figure 5—figure supplement 1F*). Importantly, this Cxcr4 induction depended on B cells, as Cxcr4 expression levels did not change in neutrophils isolated from LPS pre-exposed J$_H$T mice (*Figure 5D* and *Figure 5—figure supplement 1F*). Based on these findings and the recent observation that *Cxcr4* deficient neutrophils promote aging and neutrophil-induced vascular damage (*Adrover et al., 2019*), we hypothesized that B cells impact the life cycle of neutrophils by influencing neutrophil Cxcr4 signaling, which in turn might promote tissue damage control during a subsequent sepsis. We therefore tested whether targeting Cxcr4 would be sufficient to induce disease tolerance during sepsis and treated mice with increasing doses of the Cxcr4 pepducin agonist ATI2341, or a well-established dose of the Cxcr4 antagonist AMD3100 (*Figure 5E*). Strikingly, administration of the Cxcr4 agonist ATI2341 prevented sepsis-induced tissue damage in a dose dependent manner, whereas blocking Cxcr4 had no impact (*Figure 5F*). Liver histology reflected the tissue protective effects of ATI2341 treatment, while control and ADM3100 treated mice developed profound liver necrosis (*Figure 5G*). At the same time, none of these treatments altered the bacterial load (*Figure 5H*), suggesting that activation of Cxcr4 during sepsis induced disease tolerance.

Taken together, by studying a model of disease tolerance during sepsis, we here revealed a crosstalk between neutrophils and B cells in the bone marrow, in which B cells influence neutrophils likely by modulating Cxcr4 related pathways. In line with this idea, we found that administration of a Cxcr4 agonist improved tissue damage control during severe sepsis, indicating that Cxcr4 signaling restrains the tissue damaging properties of neutrophils during infection.

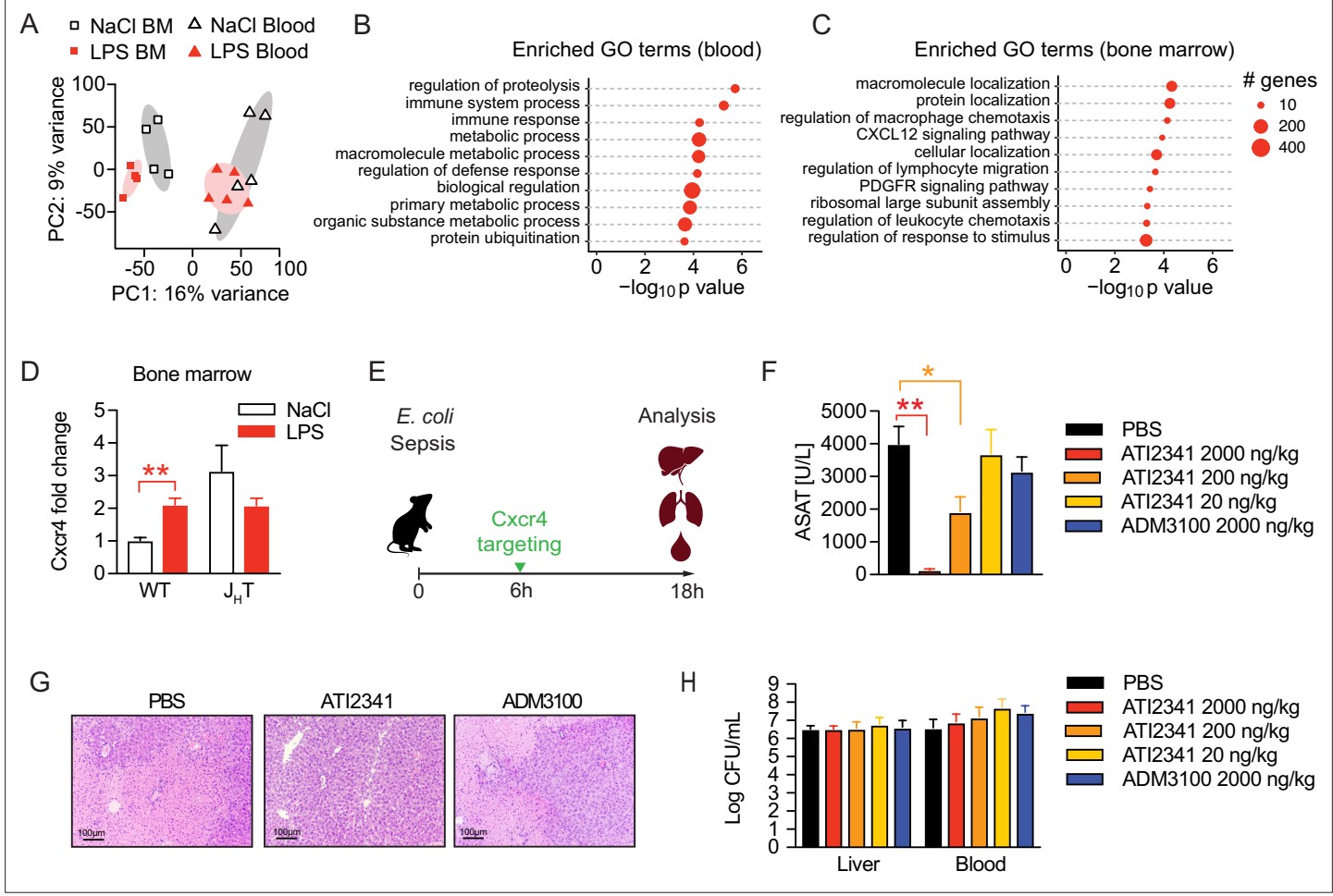

**Figure 5.** Neutrophil tissue damaging properties are modulated by bone marrow B cells via Cxcr4. (**A**) Principle componen analysis (PCA) of the top 1000 most variable genes expressed by neutrophils isolated from blood or bone marrow, of mice pretreated with lipopolysaccharide (LPS) or NaCl 2 weeks earlier. (**B–C**) Gene ontology (GO) enrichment analysis of blood and bone marrow neutrophil differentially expressed genes (DEGs). (**D**) *Cxcr4* mRNA expression in sorted bone marrow neutrophils from WT and $J_H$T mice, pretreated with NaCl or LPS 2 weeks earlier. (**E**) Schematic depiction of the treatment procedure for the therapeutic application of a Cxcr4 agonist (ATI2341) and a Cxcr4 antagonist (ADM3100) at indicated doses. (**F**) Aspartate aminotransferase (ASAT) plasma levels of mice 18 hr p.i. with *E. coli*, which were treated with depicted doses of Cxcr4 ligands (agonist ATI2341 or antagonist ADM3100, respectively) 6 hr p.i. (**G**) Representative liver histology (H&E stain) of mice 18 hr p.i. with *E. coli,* treated with PBS or the indicated Cxcr4 ligands (agonist ATI2341, antagonist ADM3100, at 2000 ng/kg) 6 hr p.i. (**H**) Liver and blood CFUs of mice 18 hr p.i. with *E. coli* of mice, which were i.v. treated with depicted doses of Cxcr4 ligands (agonist ATI2341 and antagonist ADM3100, respectively) at 6 hr p.i. Data in (**A–C**) are from a single experiment (n=4–5/group). Data in (**D**) are from an independent experiment (n=3–4), versus data shown in (**A–C**). Data in (**F–H**) are representative of two independent experiments (n=3–8/experimental group). All data are and presented as mean +/-SEM. * p≤0.05 and ** p≤0.01.

The online version of this article includes the following figure supplement(s) for figure 5:

**Figure supplement 1.** Neutrophil tissue damaging properties are modulated by bone marrow B cells via Cxcr4.

## Discussion

Sepsis-induced tissue and organ damage are severe clinical complications responsible for the high fatality rate in patients suffering from sepsis (*Mayr et al., 2014*; *Singer et al., 2016*). To this end, no therapeutic option exists that can successfully prevent organ failure in septic patients. We established and investigated a model of disease tolerance during sepsis, which enabled us to reveal the importance of B cells and neutrophils in mediating tissue damage control in the context of severe infections. Building on the reported interplay between these two cell populations, our data suggest that B cells shape neutrophils' tissue-damaging properties by modulation of neutrophil Cxcr4 signaling. Targeting Cxcr4 using a pepducin agonist protected mice from tissue damage during sepsis without affecting the bacterial load, indicating a Cxcr4-dependent disease tolerance mechanism.

Cellular depletion strategies allowed us to distinguish the contributions of selected cell types to infection-associated tissue damage in sepsis and, in parallel, to study their involvement in disease tolerance mechanisms. While we observe features of classical LPS-tolerance in our experimental setup, i.e., a reduction in early proinflammatory cytokine production, our data collectively suggest that this cannot be the sole reason for improved tissue damage control later. First, the depletion of monocytes and macrophages, which are the typical mediators of LPS-tolerance (*Fan and Cook, 2004*; *Divangahi et al., 2021*; *Freudenberg and Galanos, 1988*), did not affect liver damage in our model and did not abrogate LPS-induced protection. Second, lymphocyte deficiency was tissue protective in naïve mice without affecting cytokine levels, thus uncoupling early inflammation from tissue damage control in our model. Interestingly, the depletion of neutrophils as well as the absence of B cells fully abrogated tissue damage during primary sepsis (i.e. without prior tolerance-induction by LPS exposure), pointing towards a common, deleterious axis of these two immune cell types. It is well established that protective neutrophil effector functions during infection can be accompanied by severe collateral damage due to their tissue damaging properties by releasing inflammatory mediators such as IL-1β (*Liu and Sun, 2019*) and reactive oxygen species (*Kolaczkowska and Kubes, 2013*) or via tissue-factor mediated activation of coagulation (*Maugeri and Manfredi, 2015*; *Østerud, 2010*; *Pawlinski and Mackman, 2010*) and the release of neutrophil extracellular traps (NETs) (*Kimball et al., 2016*; *Yipp and Kubes, 2013*). Our data indicate, that neutrophils are the primary effector cells that drive tissue damage, while B cells impact tissue damage by modulating neutrophil effector functions. It was demonstrated earlier that mature, splenic B2 cells promote neutrophil activation by boosting type-I interferon (IFN) dependent early inflammation, which in turn improves bacterial clearance and survival during CLP (*Kelly-Scumpia et al., 2011*). While enhanced inflammation can mediate pathogen clearance during CLP, it at the same time contributes to tissue damage which is of particular importance in our model. In support of proinflammatory, tissue-damaging properties of mature B2 cell subsets, we found splenectomy similarly protective as B cell deficiency during primary sepsis and reconstitution of *Rag2*<sup>-/-</sup> mice with B cells to increase tissue damage. Interestingly, we did not identify an important role for the proposed IFNAR-driven inflammatory function of B cells (*Kelly-Scumpia et al., 2011*) in sepsis, and inflammation did not differ between wild type and lymphocyte deficient mice. However, it seemed counterintuitive at first, that the absence of neutrophils or B cells, respectively, prevented tissue damage in a primary infection, while they at the same time seemed critical for tissue protection in a model of LPS-induced tolerance. We thus hypothesized that B1 and B1-like cells, in contrast to B2 cells, reduced neutrophil's tissue damaging effector functions. Using soluble IgM (sIgM) deficient mice (*Ighm*<sup>-/-</sup>), enabled us to rule out a major role for IgM in tissue damage control during sepsis, even though IgM was reported to exhibit anti-thrombotic functions in cardiovascular diseases (*Binder et al., 2016*) and high plasma IgM levels positively correlate with a better outcome in human sepsis (*Krautz et al., 2018*) and mouse models (*Márquez-Velasco et al., 2007*). However, while sIgM deficiency did not prevent LPS-induced tolerance, naïve *Ighm*<sup>-/-</sup> mice developed less organ damage during primary sepsis as compared to control animals. As sIgM deficiency goes along with a decreased abundance of B2 and an increased abundance of B1 cells (*Tsiantoulas et al., 2017*) this further supported the notion of tissue damaging B2, and tissue protective B1 cells.

Since we discovered that LPS-induced protection was still observed in splenectomized animals, we considered the possibility that B cells regulate infection-induced neutrophil functionalities via effects exerted by sharing the same bone marrow niche. In fact, B cells, neutrophils and their precursors build up the majority of the constitutive CD45<sup>+</sup> bone marrow cell pool, where they mature while sharing the same niche (*Yang et al., 2013*). Due to their potential tissue damaging properties, granulopoiesis and neutrophil trafficking is tightly controlled. Under steady state conditions in mice, only 2% of mature neutrophils circulate through the body, while the majority of cells is stored in the bone marrow from where they can be quickly released upon e.g., infection, to traffic to the periphery (*Semerad et al., 2002*). Accumulating evidence highlights the substantial plasticity and functional heterogeneity of neutrophils, dependent on their localization (*Deniset et al., 2017*), circadian rhythm (*Adrover et al., 2019*), or maturation stage (*Evrard et al., 2018*).

It has been proposed earlier that neutrophils and B cells regulate each other in a reciprocal fashion in the bone marrow (*Ueda et al., 2005*). Based on our finding of a long-lasting transcriptional reprogramming in the neutrophil compartment and B cell dependency of tolerance-associated Cxcr4 upregulation, it is tempting to speculate that B cells act as important regulators of granulopoiesis

and neutrophil trafficking at steady state and under inflammatory conditions. Cxcr4 interaction with its ligand Cxcl12 (stromal cell-derived factor 1, SDF1) has been shown to be critical for the retention of neutrophils in the bone marrow under steady state, their release to the periphery as well as their homing back to the bone marrow when they become senescent (*Eash et al., 2010*; *Martin et al., 2003*). Importantly, Cxcr4 signaling is essential, as *Cxcr4* knockout mice die perinatally due to severe developmental defects ranging from virtually absent myelopoiesis and impaired B lymphopoiesis to abnormal brain development (*Ma et al., 1998*). A different sensitivity to changes in SDF1 concentrations as a potential mechanism of the reciprocal regulation of lymphopoiesis and granulopoiesis has been suggested earlier (*Ueda et al., 2005*). Antagonizing SDF1/Cxcr4 signaling is approved for stem cell mobilization from the bone marrow and is under extensive research in oncology, as it is critical for tumor development, metastasis and tumor cell migration (*Eckert et al., 2018*). More recently, Cxcr4 signaling was described to delay neutrophil aging and to protect from vascular damage in an ischemia reperfusion model (*Adrover et al., 2019*), supporting our data showing the tissue protective effects of upregulated Cxcr4 on neutrophils in sepsis. Given its clinical importance, Cxcr4 inhibition (using AMD3100) has been studied in different injury models, but interestingly only little is known about the therapeutic impact of Cxcr4 activation. Strikingly, activating, but not antagonizing, Cxcr4 during sepsis promoted tissue damage control in our model, which is in conflict with a study showing that Cxcr4 blockade with AMD3100 prior induction of peritonitis prevents neutrophil infiltration and tissue inflammation (*Ngamsri et al., 2020*). While we only see a tissue protective effect of ATI2341, but not AMD3100, we believe that this is due to differences in the timing and maybe also the route of drug administration. As we use a therapeutic approach and target Cxcr4 as late as 6 hr post *E. coli* injection, a time when there is already substantial neutrophilia in blood and organs, our data support an impact of Cxcr4 signaling on neutrophil tissue damaging properties and suggest that B cell driven regulation of Cxcr4 is a potential mechanism of disease tolerance and thus might be an interesting therapeutic target during severe sepsis.

## Materials and methods
### Animal studies
All experiments were performed using age-matched 8–12 week-old female mice. Wild type C57BL/6 mice were obtained from Charles River Laboratories or bred in the animal facility of the Medical University of Vienna. *Nfkb1*[-/-] mice were kindly provided by Derek Mann (Newcastle University, UK). *Rag2*[-/-], $J_H$T, *Ighm*[-/-], UBI-GFP, and *Ifnar1*[-/-] mice were bred in the animal facility of the Medical University of Vienna. All in vivo experiments were performed after approval by the institutional review board of the Austrian Ministry of Sciences and the Medical University of Vienna (BMWF-66.009/0272-II/3b/2013 and BMWF-66.009/0032 V/3b/2019).

### Mouse model of tolerance to *E. coli* peritonitis
Tolerance was induced by i.v. injection of 30 µg *E. coli* LPS (Sigma-Aldrich) at indicated times before induction of bacterial sepsis by intraperitoneal infection with 1–2×10⁴ *E. coli* O18:K1. *E. coli* peritonitis was induced as described previously (*Gawish et al., 2015*; *Knapp et al., 2007*; *Knapp et al., 2003*). Mice were humanely killed at indicated time points and blood, PLF and organs were collected for further analysis. Peritoneal cell numbers were determined with a hemocytometer, and cytospin preparations were stained with Giemsa for differential cell counts and/or flow cytometry. Organs were stored in 7.5% formalin for histology or homogenized using Precellys 24 (Peqlab Biotechnologie GmbH) and prepared for further analysis as described earlier in detail (*Sharif et al., 2013*). For ELISA, lysates were incubated in Greenberger lysis buffer (300 mMol NaCl, 30 mMol Tris, 2 mMol MgCl₂, 2 mMol CaCl₂, 1% Triton X-100, 2% protease inhibitor cocktail) (*Knapp et al., 2004*), and supernatants were stored at −20°C. For RNA isolation, lysates were stored in RNeasy lysis (RLT) buffer (Qiagen, containing 1% β-mercaptoethanol) at −80°C. Pathogen burden was evaluated in organ homogenates by plating serial dilutions on blood agar plates (Biomerieux), as previously described (*Gawish et al., 2015*). Blood platelet counts were determined in freshly isolated anticoagulated EDTA blood using a VetABC differential blood cell counter. Liver transaminase levels (ASAT, ALAT) were quantified in the plasma using a Cobas c311 analyzer (Roche). IL-1, IL-6, TNF, CCL2, and CXCL1 were quantified using commercial ELISA kits according to manufacturers' instructions. IgM levels were detected by coating

plates with an anti-mouse IgM capture antibody (Sigma-Aldrich), followed by blocking with 1% BSA in PBS (containing 0,27 mM EDTA) and incubation with plasma samples and standard dilutions of control mouse IgM (BioLegend). After several washing steps with PBS/EDTA, plates were incubated with an alkaline phosphatase-conjugated goat anti-mouse IgM (Sigma), washed with TBS (pH 7,4) and chemiluminescence was developed using Lumi Phos Plus (Lumigen) reagent.

## Cell depletions

Neutrophil or platelet depletion was achieved by i.v. injection of depletion antibodies 36 hr prior induction of *E. coli* peritonitis. Neutrophils were targeted using ultra-LEAF anti-Ly-6G antibody (1 mg/mouse, Biolegend) and platelets by injection of anti-GPIbα (CD42b, 40 µg/mouse) (Emfret). CD4[+] and CD8[+] T cell depletion was performed by i.v. administration of anti-CD4 (200 µg/mouse) or anti-CD8 (400 µg/mouse) antibodies 36 hr prior LPS treatment and repeated every 3 days until sepsis was induced by *E. coli* injection. Monocytes and macrophages were depleted by single i.v. administration of clodronate loaded liposomes. Depletion of platelets, T- and B cells was verified by flow-cytometry. Neutrophil depletion was confirmed by differential cell counts of Giemsa stained PLF cytospins and macrophage depletion by immunohistochemistry for F4/80[+] cells on formalin fixed liver sections.

## Cell transfers and splenectomy

Splenocytes were isolated from naïve WT C57BL/6 mice and i.v. injected into *Rag2* deficient mice (1x10[7] cells/mouse) after erythrocyte lysis using ACK lysis buffer (150 mM $NH_4Cl$, 10 mM $KHCO_3$, 0.1 mM $Na_2EDTA$, pH 7.2–7.4). Four days later, transplanted animals were pretreated with NaCl or LPS and two weeks later, challenged with *E. coli* as described above. Resting B cells were isolated from spleens of naïve UBI-GFP mice using magnetic beads (Milteny Biotec, Mouse B cell isolation kit) and i.v. injected into *Rag2* deficient mice (5x10[6] cells/mouse) after erythrocyte lysis (ACK lysis buffer) two weeks and 4 days prior to LPS/NaCl treatment. After pretreatment with NaCl or LPS transplanted animals were challenged with *E. coli* as described above. Mice were splenectomized or sham operated as described previously (*Frey et al., 2014*) and after 1 week recovery, treated with NaCl/LPS and challenged with *E. coli* as described above.

## In vitro thrombin-generation assay

Thrombin generation was assayed according to the manufacturer's instruction (Technoclone). Briefly, citrated platelet poor plasma was thawed and shortly vortexed, diluted 1:2 with PBS and transferred onto a black NUNC Maxisorp plate. Fluorogenic thrombin generation substrate containing 15 mM $CaCl_2$ was added and the plate immediately read for 60 min with Excitation/Emission at 360 nm/460 nm. Values were automatically calculated by the provided software.

## Flow-cytometry

Splenocytes were isolated by passaging spleens through 70 µm cell strainers and after erythrocyte lysis, single-cell suspensions were obtained. Bone marrow cells were obtained by flushing femurs, followed by filtering through 70 µm cell strainers. Cells were counted using a CASY cell counter and after unspecific binding was blocked using mouse IgG (Invitrogen), cells were stained in PBS containing 2% fetal calf serum (FCS) using antibodies (see table) against mouse CD45, CD3, CD19, CD23, IgM, CD21, CD43, CD11b, and Ly-6G. This was followed by incubation with a Fixable Viability Dye eFluor 780 (eBioscience) according to the manufacturer's instructions to determine cell viability. After several washing steps, cells were fixed (An der Grub Fix A reagent) and analyzed via flow cytometry using a BD LSRFortessa X-20 cell analyzer.

## Cell sorting, RNA sequencing and RT-PCR

For RNA sequencing, 200 neutrophils (defined as single/live/CD45[+]/CD3[-]/CD19[-]/Ly6G[+]/Ly-6C[int+]) were sorted from mouse bone marrow single cell suspensions (prepared as indicated above) into 4 µL cell lysis buffer containing nuclease-free $H_2O$ with 0.2% Triton X-100 (Sigma-Aldrich) and 2 U/µl RNase Inhibitor (Takara/Clonentech) using a FACSAria Fusion cytometer. Cell lysates were stored at −80°C. Library preparation was performed according to the Smart-Seq2 protocol (*Picelli et al., 2014*), followed by sequencing of pooled libraries on the Illumina HiSeq 2000/2500 (50 bp single-read setup) at the Biomedical Sequencing Facility of the Medical University of Vienna and CeMM.

For analysis, reads were adapter-trimmed (Trimmomatic) (*Bolger et al., 2014*) and aligned to the *mm10* reference genome (*STAR aligner*) (*Dobin et al., 2013*). Counting of reads mapping to genes was performed using the *summarizeOverlaps* function (*Bioconductor* R package *GenomicAlignments*) (*Lawrence et al., 2013*). Differentially expressed genes were identified using *DESeq2* (*Love et al., 2014*), whereby separate models per organ and condition (bone marrow or blood, respectively,+/-LPS or NaCl, respectively treatment) were formulated for all pairwise comparisons. Filtering was performed by independent hypothesis weighting (*ihw* R package) (*Ignatiadis et al., 2016*). GO enrichment analysis of neutrophil DEGs was performed using the GOrilla (Gene ontology enrichment analysis and visualization tool). Neutrophil sequencing data are available at the NCBI gene expression omnibus (GSE210207).

For verification of *Cxcr4* upregulation, $1.5 \times 10^5$ neutrophils were sorted as described above into cold PBS containing 2% bovine serum albumin (BSA), followed by centrifugation and resuspension of the pellet in 350 µl RLT buffer containing 1% (v/v) ß-mercaptoethanol. RNA isolation was performed using RNeasy plus micro kit (Qiagen) according to the manufacturer's instructions. Reverse transcription was performed using 150 ng of isolated RNA and the iScript cDNA synthesis kit (Biorad), according to manufacturer's instructions. Real-time PCR for mouse *Cxcr4* was performed with iTaq universal SYBR green supermix reagents (Applied Biosystems) on a StepOnePlus real-time PCR system (applied biosystems) using *Gapdh* as a housekeeper.

## Histology

Liver sections (4 µm) were stained with H&E and analyzed by a trained pathologist in a blinded fashion according to a scoring scheme, involving necrosis, sinusoidal-and lobular inflammation, steatosis, and endothelial inflammation (0 representing absent, 1 mild, 2 moderate, and 3 severe). The sum of all parameters indicated the total histology score. After staining for fresh fibrin (MSB stain, performed at the routine laboratory at Newcastle University), samples were scored for the presence of micro-thrombi by a trained pathologist in a blinded fashion. NIMPR1 immunostaining was performed on paraffin-embedded liver sections as described earlier (*Gieling et al., 2010*). Briefly, antigen retrieval was achieved using a citrate-based buffer at pH 6.0 (Vector laboratories), followed by several blocking steps. Incubation with anti-NIMP-R14 antibody (Abcam) was performed at 4°C, over-night followed by 2 hr exposure to a biotinylated secondary goat anti-rat antibody (Serotec/Biorad). For F4/80 staining, antigen retrieval was achieved by protease type XIV (Sigma) digestion at 37°C for 20 min, followed by several washings and blocking steps. After 1 hr incubation with an anti-F4/80 antibody (Serotec), exposure to a biotinylated secondary rabbit anti-rat antibody was performed at room temperature. Finally, both stains were incubated with VECTASTAIN Elite ABC Reagent and visualized using diaminobenzidine peroxidase substrate (Vector Laboratories) followed by counterstaining with hematoxylin and embedding (Eukitt, Sigma).

## Statistical analysis

Statistical evaluation was performed using GraphPad Prism software except for statistical analysis of RNA sequencing data, which was performed using R. Data are represented as mean ± SEM and were analyzed using either Student´s t-test, comparing two groups, or one-way ANOVA analysis, followed by Tukey multiple comparison test, for more than two groups. Differences with a p-value ≤0.05 were considered significant. For DEG, genes with an FDR-adjusted p-value of <0.1 were considered differentially expressed.

## Acknowledgements

The authors thank Henriette Luise Horstmeier and Aysu Eshref for help with experimental procedures, Sophie Zahalka for help with illustrations and the animal caretakers at the Center for Biomedical Research, Medical University Vienna, for their expert work. We acknowledge the core facilities of the Medical University of Vienna, including the Flow Cytometry Facility and the Biomedical Sequencing Facility (BSF, jointly run with CeMM). This work was funded by the Austrian Science Fund (FWF) within the Doctoral Program Cell Communication in Health and Disease (CCHD, 1205FW), and the Special Research Projects Immunothrombosis (SFB 054–10) and Chromatin Landscapes (SFB 061–04) to SK. FO is supported by MRC program Grants; MR/K0019494/1 and MR/R023026/1.

## Additional information

### Competing interests

Louis Boon: is affiliated with Polypharma Biologics. The author has no financial interests to declare. The other authors declare that no competing interests exist.

### Funding

| Funder | Grant reference number | Author |
| --- | --- | --- |
| Austrian Science Fund | CCHD | Riem Gawish |
| Austrian Science Fund | SFB 061-04 | Sylvia Knapp |
| MRC-PHE Centre for Environment and Health | MR/K0019494/1 | Fiona Oakley |
| Austrian Science Fund | 1205FW | Riem Gawish |
| MRC program Grants | MR/K0019494/1 | Fiona Oakley |

The funders had no role in study design, data collection and interpretation, or the decision to submit the work for publication.

### Author contributions

Riem Gawish, Conceptualization, Data curation, Formal analysis, Funding acquisition, Validation, Investigation, Methodology, Writing - original draft, Writing - review and editing; Barbara Maier, Conceptualization, Data curation, Formal analysis, Validation, Investigation, Methodology; Georg Obermayer, Anna-Dorothea Gorki, Federica Quattrone, Asma Farhat, Anastasiya Hladik, Ana Korosec, Arman Alimohammadi, Investigation; Martin L Watzenboeck, Data curation, Formal analysis; Karin Lakovits, Investigation, Project administration; Ildiko Mesteri, Felicitas Oberndorfer, John Brain, Formal analysis; Fiona Oakley, Conceptualization, Supervision; Louis Boon, Irene Lang, Christoph J Binder, Supervision; Sylvia Knapp, Conceptualization, Supervision, Funding acquisition, Writing - original draft, Writing - review and editing

### Author ORCIDs

Riem Gawish http://orcid.org/0000-0003-4267-2131
Anna-Dorothea Gorki http://orcid.org/0000-0002-2521-0333
Sylvia Knapp http://orcid.org/0000-0001-9016-5244

### Ethics

All in vivo experiments were performed after approval by the institutional review board of the Austrian Ministry of Sciences and the Medical University of Vienna (BMWF-66.009/0272-II/3b/2013 and BMWF-66.009/0032-V/3b/2019).

### Decision letter and Author response

Decision letter https://doi.org/10.7554/eLife.78291.sa1
Author response https://doi.org/10.7554/eLife.78291.sa2

## Additional files

### Supplementary files

• Supplementary file 1. DEG of blood neutrophils isolated from mice 2 weeks post NaCl or LPS treatment.

• Supplementary file 2. DEG of bone marrow neutrophils isolated from mice 2 weeks post NaCl or LPS treatment.

• MDAR checklist

### Data availability

Neutrophil sequencing data are deposited in NCBI Gene Expression Omnibus (GSE210207).

The following dataset was generated:

| Author(s) | Year | Dataset title | Dataset URL | Database and Identifier |
|---|---|---|---|---|
| Gawish R, Gorki A, Watzenboeck M, Farhat A, Knapp S | 2022 | A neutrophil - B-cell axis impacts tissue damage control during sepsis via Cxcr4 | https://www.ncbi.nlm.nih.gov/geo/query/acc.cgi?acc=GSE210207 | NCBI Gene Expression Omnibus, GSE210207 |

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

# Appendix 1

## Reagents and resources

### Appendix 1—key resources table

| Reagent type (species) or resource | Designation | Source or reference | Identifiers | Additional information |
|---|---|---|---|---|
| Strain, strain background (*Mus musculus*, female) | C57BL/6 J | Own colony, Jackson Labs | JAX #000664 | |
| Strain, strain background (*Mus musculus*, female) | B6;129P-Nfkb1tm1Bal/J (*Nfkb1⁻/⁻*) | Provided by Derek Mann (Newcastle University, UK), Jackson Labs | JAX #002849, (*Sha et al., 1995*) | |
| Strain, strain background (*Mus musculus*, female) | B6.Cg-Rag2tm1.1Cgn/J (*Rag2⁻/⁻*) | Own colony, Jackson Labs | JAX #008449, (*Hao and Rajewsky, 2001*) | |
| Strain, strain background (*Mus musculus*, female) | B6.129P2-Igh-Jtm1Cgn/J (J$_H$T) | Own colony, Jackson Labs | JAX #002438, (*Gu et al., 1993*) | |
| Strain, strain background (*Mus musculus*, female) | B6;129S4-Ighmtm1Che/J (*Ighm⁻/⁻*) | Own colony, Jackson Labs | JAX #003751, (*Boes et al., 1998*) | |
| Strain, strain background (*Mus musculus*, female) | B6.129S2-Ifnar1tm1Agt/Mmjax (*Ifnar1⁻/⁻*) | Own colony, Jackson Labs | MMRRC Strain #032045-JAX, (*Müller et al., 1994*) | |
| Strain, strain background (*Mus musculus*, female) | C57BL/6-Tg(UBC-GFP)30Scha/J (UBI-GFP) | Own colony, Jackson Labs | JAX #004353, (*Schaefer et al., 2001*) | |
| Strain, strain background (*E. coli*) | *E. coli*, O18:K1 | Clinical isolate | n.a. | |
| Antibody | Ultra-LEAF anti-mouse Ly-6G (rat monoclonal) | BioLegend | Cat# 127650; RRID:AB_2572002 | '1 mg/mouse' i.v. |
| Antibody | anti-mouse GPIbα (CD42b) (rat monoclonal) | Emfret | Cat# R300; RRID:AB_2721041 | '40 µg/mouse' i.v. |
| Antibody | anti-mouse CD4 (rat monoclonal) | Generated in house, clone GK1.5 | n.a. | '200 µg/mouse' i.v. |
| Antibody | anti-mouse CD8 (rat monoclonal) | Generated in house, clone YTS169 | n.a. | '400 µg/mouse' i.v. |
| Antibody | PE anti-mouse CD61 (armenian hamster monoclonal | BioLegend | Cat# 104307; RRID:AB_313084 | FC '(1:200)' |
| Antibody | BV510 anti-mouse CD45 (rat monoclonal) | BioLegend | Cat# 103138; RRID:AB_2563061 | FC '(1:200)' |
| Antibody | APC/Cy7 anti-mouse TER-119 (rat monoclonal) | BioLegend | Cat# 116223; RRID:AB_2137788 | FC '(1:200)' |
| Antibody | FITC anti-mouse CD3 (rat monoclonal) | BioLegend | Cat# 100204; RRID:AB_312661 | FC '(1:200)' |
| Antibody | PerCP/Cy5.5 anti-mouse CD4 (rat monoclonal) | BioLegend | Cat# 100433; RRID:AB_893330 | FC '(1:100)' |
| Antibody | Pacific Blue anti-mouse CD8a (rat monoclonal) | BioLegend | Cat# 100728; RRID:AB_493426 | FC '(1:200)' |
| Antibody | FITC anti-mouse CD19 (rat monoclonal) | BioLegend | Cat# 115506; RRID:AB_313641 | FC '(1:200)' |

*Appendix 1 Continued*

| Reagent type (species) or resource | Designation | Source or reference | Identifiers | Additional information |
|---|---|---|---|---|
| Antibody | BV605 anti-mouse CD19 (rat monoclonal) | BioLegend | Cat# 115540; RRID:AB_2563067 | FC '(1:200)' |
| Antibody | PE anti-mouse CD19 (rat monoclonal) | BioLegend | Cat# 115507; RRID:AB_313642 | FC '(1:200)' |
| Antibody | PE anti-mouse IgD (rat monoclonal) | BioLegend | Cat# 405706; RRID:AB_315028 | FC '(1:200)' |
| Antibody | eFluor450 anti-mouse IgM (rat monoclonal) | eBioscience, Thermo Fisher | Cat# 48-5890-80; RRID:AB_10671342 | FC '(1:200)' |
| Antibody | FITC anti-mouse CD23 (rat monoclonal) | BioLegend | Cat# 101606; RRID:AB 312831 | FC '(1:100)' |
| Antibody | PerCP/Cy5.5 anti-mouse CD21/CD35 (CR2/CR1) (rat monoclonal) | BioLegend | Cat# 123416; RRID:AB_1595490 | FC '(1:100)' |
| Antibody | APC anti-mouse CD43 (rat monoclonal) | BioLegend | Cat# 143208; RRID:AB_1114965 | FC '(1:200)' |
| Antibody | PE/Cy7 anti-mouse Ly-6G (rat monoclonal) | BioLegend | Cat# 127617; RRID:AB_1877262 | FC '(1:200)' |
| Antibody | PE anti-mouse Ly6G (rat monoclonal) | BioLegend | Cat# 127608; RRID:AB_1186099 | FC '(1:200)' |
| Antibody | Brilliant Violet 605 anti-mouse Ly-6C (rat monoclonal) | BioLegend | Cat# 128036; RRID:AB_2562353 | FC '(1:200)' |
| Antibody | Alexa Fluor 700 anti-mouse CD11b (rat monoclonal) | BioLegend | Cat# 101222; RRID:AB_493705 | FC '(1:200)' |
| Antibody | PE anti-mouse CD62L (rat monoclonal) | BD Biosciences | Cat# 553151; RRID:AB_394666 | FC '(1:200)' |
| Antibody | APC anti-mouse Cxcr4 (rat monoclonal) | BioLegend | Cat# 146507; RRID:AB_2562784 | FC '(1:200)' |
| Antibody | anti-mouse NIMP-R14 (rat monoclonal) | Abcam | Cat# ab2557-50; RRID:AB_303154 | IHC '(1:50)' |
| Antibody | Biotin anti-rat IgG (goat polyclonal) | BioRad | Cat# STAR131B; RRID:AB_11152774 | IHC '(1:200)' |
| Antibody | anti-mouse F4/80 (rat monoclonal) | AbD Serotec | Cat# MCA497G; RRID:AB_872005 | IHC '(1:200)' |
| Antibody | Biotin anti-rat IgG (goat polyclonal) | Vector Laboratories | Cat# BA-4001; RRID:AB_10015300 | IHC '(1:200)' |
| Antibody | anti-mouse IgM (goat polyclonal) | Sigma-Aldrich | Cat# M8644; RRID:AB_260700 | ELISA '(2 µg/mL)' |
| Antibody | anti-mouse IgM, κ isotype control antibody (mouse monoclonal) | BioLegend | Cat# 401602 | ELISA '(0,781–50 ng/mL)' |
| Antibody | Alkaline phosphatase anti-mouse IgM, (goat polyclonal) | Sigma-Aldrich | Cat# A9688; RRID:AB_258472 | ELISA '(1:20000)' |
| Antibody | anti-mouse CD16/32 (rat monoclonal) | BioLegend | Cat# 101320; RRID:AB_1574975 | FC '(1:50)' |
| Sequence-based reagent | mouse *Cxcr4* fwd | This study | PCR primers, Microsynth | TGCAGCAGGTAGCAGTGAAA |
| Sequence-based reagent | mouse *Cxcr4* rev | This study | PCR primers, Microsynth | TGTATATACTCACACTGATCGGTCC |
| Sequence-based reagent | mouse *Gapdh* fwd | This study | PCR primers, Microsynth | GGTCGTATTGGGCGCCTGGTCACC |

*Appendix 1 Continued on next page*

*Appendix 1 Continued*

| Reagent type (species) or resource | Designation | Source or reference | Identifiers | Additional information |
|---|---|---|---|---|
| Sequence-based reagent | mouse *Gapdh* rev | This study | PCR primers, Microsynth | CACACCCATGACGAAC ATGGGGGC |
| Peptide, recombinant protein | Bovine serum albumin | Sigma-Aldrich | Cat# A8806 | |
| Peptide, recombinant protein | Protease Type XIV | Sigma-Aldrich | Cat# P5147 | |
| Commercial assay or kit | Mouse IL-6 ELISA | BioLegend | Cat# 431301 | |
| Commercial assay or kit | Mouse Cxcl1/KC DuoSet ELISA | R&D Systems | Cat# DY453 | |
| Commercial assay or kit | Mouse IL-1β ELISA | BioLegend | Cat# 432601 | |
| Commercial assay or kit | Mouse Ccl2/MCP-1 DuoSet ELISA | R&D Systems | Cat# DY479 | |
| Commercial assay or kit | Avidin/Biotin blocking kit | Vector Labs | Cat# SP-2001 | |
| Commercial assay or kit | Vectastain ABC kit | Vector Labs | Cat# PK-6100 | |
| Commercial assay or kit | DAB Subtrate kit | Vector Labs | Cat# SK-4100 | |
| Commercial assay or kit | RNeasy Plus micro kit | Gibco | Cat# 74034 | |
| Commercial assay or kit | iScript cDNA Synthesis kit | BioRad | Cat#170–8891 | |
| Commercial assay or kit | iTaq Universal SYBR Green Supermix | BioRad | Cat#172–5124 | |
| Commercial assay or kit | TECHNOTHROMBIN TGA Assay | Technoclone | Cat#5006010 | |
| Chemical compound, drug | Lipopolysaccharide purified from *E. coli* O55:B5 | Sigma-Aldrich | cat# L2880 | |
| Chemical compound, drug | Endotoxin-free PBS, pH 7.4 | Gibco | Cat# 11503387 | |
| Chemical compound, drug | Fixable Viability Dye eFluor 780 | ThermoFisher, eBioscience | Cat# 65-0865-14 | FC '(1:3000)' |
| Chemical compound, drug | NaCl | Carl Roth | Cat# 0601.1 | |
| Chemical compound, drug | EDTA | Sigma-Aldrich | Cat# E5134 | |
| Chemical compound, drug | TRIS | VWR Chemicals | Cat# 28808.294 | |
| Chemical compound, drug | $MgCl_2$ | Sigma Aldrich | cat# M8266 | |
| Chemical compound, drug | $CaCl_2$ | Sigma Aldrich | cat# C3306 | |

*Appendix 1 Continued on next page*

*Appendix 1 Continued*

| Reagent type (species) or resource | Designation | Source or reference | Identifiers | Additional information |
|---|---|---|---|---|
| Chemical compound, drug | Triton X-100 | Sigma-Aldrich | cat# T9284 | |
| Chemical compound, drug | NH$_4$Cl | Sigma-Aldrich | cat# 09718 | |
| Chemical compound, drug | KHCO$_3$ | Carl Roth | cat# P748.1 | |
| Chemical compound, drug | Na$_2$EDTA | Sigma-Aldrich | cat# 324503 | |
| Chemical compound, drug | RLT Plus Buffer | Qiagen | cat# 1053393 | |
| Chemical compound, drug | β-mercaptoethanol | Sigma-Aldrich | cat# M3148 | |
| Chemical compound, drug | Formalin 7.5% | SAV LP GmbH | cat# FN-60180-75-1 | |
| Chemical compound, drug | Citrate based antigen unmasking solution | Vector laboratories | cat# H3300 | |
| Chemical compound, drug | Eosin Y Solution | Sigma-Aldrich | cat# 318906 | |
| Chemical compound, drug | Hematoxylin solution (Mayer´s) | Sigma-Aldrich | Cat# MHS16 | |
| Chemical compound, drug | Mayer´s Hemalum solution | Merck | cat# 654833 | |
| Chemical compound, drug | Clodronate loaded liposomes | http://www.clodronateliposomes.org | cat# C-010 | |
| Chemical compound, drug | Protease inhibitor cocktail | Sigma-Aldrich | cat# P8340 | |
| Chemical compound, drug | RNase Inhibitor | Takara/Clonentech | cat# 2313 A | |
| Chemical compound, drug | Eukitt | Sigma-Aldrich | cat# 03989 | |
| Chemical compound, drug | Lumi Phos plus | Lumigen, Beckmann Coulter | cat# P-701 | |
| Chemical compound, drug | Antigen Unmasking Solution | Vector Labs | Cat# H3300-250 | |
| Software, algorithm | GraphPad Prism 9.1 | Graphpad Software, Inc. | https://www.graphpad.com | |
| Software, algorithm | FlowJo | Becton, Dickinson and Company | https://www.flowjo.com/ | |
| Software, algorithm | Bioconductor R package *Genomic alignments* | *Lawrence et al., 2013* | | |

*Appendix 1 Continued on next page*

*Appendix 1 Continued*

| Reagent type (species) or resource | Designation | Source or reference | Identifiers | Additional information |
|---|---|---|---|---|
| Software, algorithm | DSeq2 | *Love et al., 2014* | | |
| Software, algorithm | GOrilla | *Eden et al., 2009* | | |
| Software, algorithm | *ihw* R package | *Ignatiadis et al., 2016* | | |
| Software, algorithm | STAR aligner | *Dobin et al., 2013* | | |
| Software, algorithm | Trimmomatic | *Bolger et al., 2014* | | |
| Other | Columbia agar plates +5% sheep blood | Biomerieux | http://www.biomerieux-culturemedia.com/ | *E. coli* CFU counts |
| Other | Fetal Bovine Serum | Sigma | Cat# F9665 | Blocking and flow cytometry |
| Other | Goat Serum | Novus Biologicals | Cat# NBP2-23475 | Blocking |

