## [Editor Report]

In this elegant animal experiment on a novel mouse model, the authors' aim was to investigate the mechanisms of long-lasting disease tolerance to identify potential targets to prevent organ damage in sepsis. The results showed that interaction between B cells and neutrophils plays a pivotal role in preventing tissue damage and modulation, rather than upregulation of the Cxcr4 expression of neutrophils requires the presence of B cells, and this promotes disease tolerance by improving tissue damage control via the suppression of neutrophils' tissue damaging properties. Furthermore, treatment with a Cxcr4-agonist successfully replicated the tissue tolerance phenotype and prevented organ damage. These results could have important clinical and research implications by unveiling further Cxcr-dependent mechanisms and potentially can lead to the development of a novel therapeutic approach to sepsis.

---

## [Decision Letter]

**Decision letter after peer review:**

Thank you for submitting your article "A neutrophil -B-cell axis governs disease tolerance during sepsis via Cxcr4" for consideration by *eLife*. Your article has been reviewed by 2 peer reviewers, and the evaluation has been overseen by a Reviewing Editor and Carla Rothlin as the Senior Editor. The reviewers have opted to remain anonymous.

Essential revisions:

1) Correct the term sepsis-tolerance to LPS-tolerance in the title and also throughout the manuscript

2) Please better explain the rationale of explanation leading from B-cell modulation to Cxcr4 and neutrophil activation

3) The novelty of the paper needs to be clarified by putting the comments of Reviewer #2 into context.

*Reviewer #1 (Recommendations for the authors):*

– The authors discuss sepsis tolerance. However, the correct term is LPS tolerance. They are using the traditional LPS tolerance approach which is different from sepsis tolerance. Should they need to report on sepsis tolerance, all studied mice should have been subject to moderate bacteremia or to be survivors from CLP sepsis. This needs to be acknowledged in the Discussion and amended throughout the manuscript.

– The model the authors are using is based on survival benefit 24 hours following challenge with one small amount of LPS. Why then do the authors state that they have decided to study mice primed the last two weeks with LPS?

– I suggest that in parallel to sacrifice experiments showing the laboratory read-out of deleted elements of the immune system, some survival experiments are necessary particularly with the Rag2-/- mice.

– LPS tolerance and modulation of the neutrophil phagocytosis requires decrease of the tissue bacterial load. Figure 1A does not suggest this.

– The authors need to better explain the rationale of explanation leading from B-cell modulation to Cxcr4 and neutrophil activation.

*Reviewer #2 (Recommendations for the authors):*

Although these data could certainly fine-draw the border of our knowledge on preconditioning/tolerance and sepsis-induced PMN changes in rodents, the study is largely recombinant and repetitive and the human/translational significance is also questionable.

---

## [Author Response]

Essential revisions:1) Correct the term sepsis-tolerance to LPS-tolerance in the title and also throughout the manuscript

We thank the reviewers for this suggestion and apologize for the confusion we might have created by not sufficiently explaining the terminology used in our manuscript and by not explicitly differentiating between “LPS-tolerance” and “disease tolerance”.

As legitimately stated by the reviewers, LPS-tolerance has been excessively studied many years ago and describes the phenomenon of reduced inflammatory responses to TLR stimulation of (myeloid) cells which have been previously exposed to LPS (Zeisberger and Roth 1998, Fan and Cook 2004). While this definition of LPS-tolerance is still valid in vitro, the term was subsequently used in multiple papers for all types of in vivo setups, where an initial LPS exposure protected from ANY second insult without investigating the underlying cellular and molecular mechanisms of protection. According to more recent literature, LPS tolerance is considered as a specific type of innate immune cell memory (Divangahi, Aaby et al., 2021) and is no longer the adequate term for many of those in vivo experimental setups, including the experimental setup we describe in our manuscript, which is why we have chosen the term “disease tolerance”.

“LPS tolerance” is today seen as a type of innate memory. It is typically mediated via monocytes and macrophages and was suggested to depend on the upregulation of negative regulators of TLR signaling, such as IRAK-M and the formation of inhibitory NFκB p50 homodimers (Fan and Cook 2004). While LPS tolerance (and other innate immune cell states like “differentiation”, “priming” or “training”, as reviewed in (Divangahi, Aaby et al., 2021)) can be nicely differentiated from each other in vitro, the clean separation of these effects in vivo is often impossible. With regards to LPS-tolerance, the in vitro phenomenon of a reduced cytokine response could be recapitulated as an outcome-driving mechanism in rather simple in vivo setups. Mice pretreated with either single or repetitive doses of LPS were protected from a final, otherwise fatal hit with LPS due to a suppressed cytokine storm (Zeisberger and Roth 1998) and protected from liver damage in a hepatic ischemia reperfusion model, again by reducing inflammatory cytokine-induced tissue injury (Sano, Izuishi et al., 2010). In line with established in vitro mechanisms, the importance of monocytes and macrophages for LPS tolerance in these in vivo setups could be elegantly shown by cell transfer experiments (Freudenberg and Galanos 1988). However, the term LPS-tolerance might not be 100% appropriate in cases where the second hit is a replicating microbiological agent. For example, repetitive oral LPS administration has been shown to protect mice in a subsequent cecal ligation and puncture (CLP) sepsis model (Marquez-Velasco, Masso et al., 2007). While the authors show that LPS-preconditioned mice exhibit lower cytokine levels in late stages of CLP, it remains unclear whether this is due to real LPS-tolerance of innate immune cells (i.e. a reduced responsiveness of cells) or whether this is the result of a reduced bacterial burden which would mean less stimulus exposure and subsequently less cytokine production. To make it even more complicated, a reduced bacterial burden could even be the consequence of a strong and powerful early cytokine response (due to innate immune training, the opposite of LPS-tolerance), which would then result in an improved bacterial clearance and better outcome upon CLP later. In line with this, we have just recently published that intranasal LPS pretreatment protects mice in a subsequent sepsis induced by S. Pneumoniae. Again, the protective mechanism was not “LPS tolerance”, but “training” which resulted in an improved clearance of bacteria (Zahalka, Starkl et al., 2022).

The term “disease tolerance”, which we used in our manuscript, was originally discovered in plant biology (Schafer 1971) and was only recently introduced to animals and human. In contrast to LPS-tolerance, “disease tolerance” goes beyond the single cell layer and was promoted a few years ago by several groups (most prominently by the groups of Ruslan Medzhitov, Miguel Soares and Janelle Ayres) as a new concept of host defense (Medzhitov, Schneider et al., 2012, McCarville and Ayres 2018, Martins, Carlos et al., 2019). The core concept of disease tolerance can probably be considered an extension of the earlier ideas of LPS tolerance and describes the phenomenon that some immune response pathways are not aimed at the elimination of microbes, but instead meant to protect the host from disease for example by maintaining tissue integrity during infection, irrespective of the pathogen burden. In other words, as explained by Janelle Ayres: “Two distinct defense strategies provide a host with survival to infectious diseases: resistance and tolerance. Resistance is dependent on the ability of the host to kill pathogens. Tolerance promotes host health while having a neutral to positive impact of pathogen fitness.” As such, two hosts with the same pathogen load can either get sick or stay relatively healthy (i.e. tolerant).

Mechanistically, tissue damage control, a terminology which we are now using in our revised manuscript instead of tissue tolerance, underlies many protective effects of disease tolerance and the underlying players that enable disease tolerance can vary widely and involve metabolic, inflammatory, innate or adaptive immunity, and even structural cells or neurons, to name a few (McCarville and Ayres 2018). In addition, any external trigger, be it microbial (e.g. LPS or others), metabolites or drugs, can potentially interfere with players of disease tolerance, thereby enabling or preventing it.

Importantly, most human and mouse studies in the field of severe infection and sepsis have focused on resistance mechanisms and antimicrobial therapies have been successfully developed to support pathogen clearance, which is why disease tolerance can be called a neglected defense strategy, as was just recently highlighted (“Immunology´s intolerance of disease tolerance” (Schneider 2021)).

Severe sepsis patients, in particular, often die due to tissue damage and subsequent organ failure in consequence of a derailed immune response to infection – despite of proper control of the pathogen load (by the use of antibiotics). It is not well understood why some patients can maintain organ integrity and survive while others slip into a fatal circle of dysregulated inflammation and coagulation, and finally die from organ failure. Importantly, the immunopathology of severe sepsis cannot be simply explained by uncontrolled cytokine responses by monocytes or macrophages but is caused by a derailed network of interacting cells. We therefore are convinced that a better understanding of cellular and molecular disease tolerance mechanisms can help to identify targets for reduced sepsis mortality. This is our motivation for the submitted work.

In this revision, we attempted to:

i) better explain the difference and overlaps of LPS-tolerance and disease tolerance in the introduction of the manuscript, and ii) provide a list of explanations and additions to support our claim of disease tolerance (and not exclusively LPS tolerance).

ii) better explanation of the differences between LPS-tolerance and disease tolerance: We are now using the established term “tissue damage control” as an underlying cause for the protective impact of “disease tolerance” and removed the term “tissue tolerance” from our manuscript and in detail discuss how LPS-tolerance is involved in our phenotype.

We have adjusted the title to avoid the term tissue tolerance:

“… B-cell axis impacts tissue damage control during sepsis via Cxcr4”. page 1, line 1

We added the following statements to abstract and introduction in hopes this will help to better understand the context of our study and the terminology we have used:

“Neutrophil Cxcr4 upregulation required the presence of B cells, suggesting that B cells promoted disease tolerance by improving tissue damage control via the suppression of neutrophils’ tissue damaging properties. Finally, therapeutic administration of a Cxcr4 agonist successfully promoted tissue damage control and prevented liver damage during sepsis…” Page 2, line 8-12

“At the same time, sepsis goes along with a state of immune cell dysfunction characterized by immune cell exhaustion, enhanced apoptosis and impaired antigen presentation, cytokine production and pathogen killing (Hotchkiss, Monneret et al., 2013).” Page 3, lines 10-12

“The potent immunogen lipopolysaccharide (LPS) is an important driver of inflammation during Gram-negative sepsis, due to its ability to activate Toll-like receptor (TLR) 4 (Hoshino, Takeuchi et al., 1999, Beutler 2000) and caspase-11 (Hagar, Powell et al., 2013, Kayagaki, Wong et al., 2013). The state of sepsis-induced immune paralysis can be partially recapitulated in vitro by the phenomenon of LPS tolerance, which is characterized by an unresponsive state of myeloid cells after repeated TLR stimulation (Fan and Cook 2004, de Vos, Pater et al., 2009). In contrast, the concept of “disease tolerance” describes a poorly studied, yet essential host defense strategy, on top of the well understood strategies of avoidance and resistance (Medzhitov, Schneider et al., 2012). While avoidance means preventing pathogen exposure and infection, and resistance aims to more efficiently reduce the pathogen load in the course of an established infection, disease tolerance involves mechanisms, which minimize the detrimental impact of infection irrespective of the pathogen burden, thus improving host fitness despite the infection for example by improving tissue damage control (Medzhitov, Schneider et al. 2012, Martins, Carlos et al., 2019). Importantly, LPS-tolerance can impact both, pathogen resistance and/or disease tolerance.” Page 3, lines 15-26

“In this study, we investigated mechanisms of disease tolerance and tissue damage control, by comparing tolerant and sensitive hosts during a severe bacterial infection. While sensitive animals developed severe coagulopathy and tissue damage during sepsis, tolerant animals showed improved tissue damage control and were able to maintain tissue integrity in spite of a high bacterial load. Disease tolerance was induced by the prior exposure of animals to a single, low-dose of LPS and could be uncoupled from LPS-induced suppression of cytokine responses.” Page 3, lines 32-34, Page 4, lines 1-3

“… strategy to foster tissue damage control in severe sepsis.” Page 4, lines 11

iii) explanations and additions to support our claim of disease tolerance:

Taking past and recent literature into consideration, we argue that the herein studied tissue protection during sepsis is in fact improved tissue damage control, which is one key aspect of “disease tolerance”. In contrast, the term “LPS-tolerance” would not cover the full spectrum of effects we observe.

a) All in vivo studies about LPS-tolerance either used repetitive LPS preconditioning or a very short time window of 1-2 days between first and second hit and no studies exist, which address the long-term effects of LPS exposure on a subsequent lethal infection. Our model shows that a single subclinical LPS dose 2 days prior bacterial infection improved bacterial clearance and tissue protection (a similar setup like in other studies, which would not allow to specifically study tissue damage control mechanisms due to differences in the pathogen load) (Figure 1A and 1B). Importantly, when LPS was administered 2, 5 or 8 weeks prior bacterial infection bacterial clearance was no longer improved (Figure 1A), whereas organ-protection during sepsis was maintained (Figure 1B). This setting is defined as “disease tolerance” and was used for all follow-up experiments.

We explicitly address this in our manuscript with the following statements:

“LPS pre-exposure has been shown to prevent hyperinflammation during fatal endotoxemia via the induction of LPS-tolerance (Lopez-Collazo and del Fresno 2013) and to improve the outcome of CLP sepsis by promoting bacterial clearance (MarquezVelasco, Masso et al., 2007). To investigate mechanisms of tissue damage control and disease tolerance during bacterial sepsis we aimed for a setup with a stable pathogen load which would allow us to only interfere with sepsis-associated organ failure. We thus challenged mice intravenously (i.v.) with a subclinical dose of LPS 1 day, 2 weeks, 5 weeks or 8 weeks, respectively, prior to the induction of Gram-negative sepsis by intraperitoneal (i.p.) injection of the virulent *E. coli* strain O18:K1.” Page 5, lines 3-9

“Thus, short-term (24h) LPS pre-exposure improved resistance to infection and consequently tissue integrity, while long-term (2-8 weeks) LPS pre-exposure enabled the maintenance of tissue integrity irrespective of a high bacterial load, which per definition resembles disease tolerance. To dissect the underlying mechanism of tissue damage control in disease tolerant mice, we thus performed all subsequent experiments by treating mice with either LPS or saline two weeks prior to bacterial infection, allowing us to compare tolerant with sensitive hosts.” Page 5, lines 15-21

b) LPS-tolerance has been attributed to myeloid cells, typically monocyte and macrophage reprogramming and diminished cytokine responses (Freudenberg and Galanos 1988, Fan and Cook 2004). If LPS-tolerance was the main reason for organ protection in our model, one would assume that the absence of “tolerized” monocytes/macrophages would result in aggravated liver tissue damage. In our setup, the depletion of monocytes and macrophages did neither affect organ damage parameters in control mice (i.e. those cells do not contribute to organ damage) nor tolerant mice (i.e. those cells do not mediate tolerance) (Figure 4A). Moreover, IL-6 levels in PLF of chlodronate treated animals were substantially reduced (data not shown in manuscript), while liver IL-6 remained unaffected by MonoMac depletion, suggesting that i) LPS-tolerance in monocytes and macrophages was not driving disease tolerance in our model and ii) reduced cytokine responses were not directly coupled to organ damage.

We have adapted the following statement in the result section of our manuscript to clarify this point:

Surprisingly, monocyte and macrophage depletion neither influenced sepsis-induced tissue damage in sensitive animals nor did it impact on LPS-induced tissue damage control (Figure 4A), suggesting that classical LPS-tolerance is not the sole reason for protection. Page 8, lines 23-26

And discuss it in more detail in the discussion:

“While we observe features of classical LPS-tolerance in our experimental setup, i.e. a reduction in early proinflammatory cytokine production, our data collectively suggest that this cannot be the sole reason for improved tissue damage control later. First, the depletion of monocytes and macrophages, which are the typical mediators of LPS-tolerance (Freudenberg and Galanos 1988, Fan and Cook 2004, Divangahi, Aaby et al., 2021), did not affect liver damage in our model and did not abrogate LPS-induced protection.” Page 12, lines 14-18

c) While we observe aspects reminiscent of in vivo LPS-tolerance (i.e. reduced early (6h) cytokine response in LPS-pre-exposed animals as shown in Figure 2H and Supp. Figure 2G), our data suggest that this cannot explain improved tissue damage control later. First, cytokine levels at the time when first signs of sepsis-induced liver damage are detectable (which is only 16 – 18 h after infection), are indistinguishable between naïve and LPS pretreated mice (Figure 1F and Supp. Figure 1B-C). Second, while cytokine storm is typically considered associated with sepsis-induced organ damage, mice deficient for B- and T cells (Rag2^-/-^ mice) are clearly less prone to liver damage (Figure 2A), but show similar levels of IL-6, CXCL1 and CCL2 (vs. wild type mice) (Figure 2H-I and Supp. Figure 2G-H). These findings uncouple inflammatory cytokine levels from organ damage in our model and suggest other mechanisms to be involved in the development of organ failure. Third, and most importantly, LPS pretreatment suppressed early inflammation in animals deficient for B and T cells (Figure 2I and Supp. Figure 2H), but did not lead to less severe liver damage during subsequent *E. coli* sepsis (Figure 2A), thus uncoupling early cytokine levels from organ damage in our setting.

We explain this in our Results section and have now slightly modified the text to clarify this point better:

“Six hours post *E. coli* infection, we found tolerant wild type mice to exhibit lower IL-6 levels in blood and liver (Figure 2H), as well as lower amounts of important regulators of peritoneal leukocyte migration (Bianconi, Sahebkar et al., 2018, Rajarathnam, Schnoor et al., 2019), like CXCL1 and CCL2 (Figure 2—figure supplement 1G) when compared to sensitive control mice, a phenotype which is reminiscent of LPS-tolerance. However, lymphocyte deficient Rag2-/- animals, in whom tissue damage control could not be improved by LPS preexposure (Figure 2A and Figure 2—figure supplement 1A), showed comparable reductions in these mediators of early inflammation in response to LPS pretreatment (Figure 2I and Figure 2—figure supplement 1H).” page 7, line 8-15

“These data suggested that in tolerant hosts, B cells contributed to tissue protection during sepsis, and that an LPS mediated modulation of early inflammation is unlikely to explain these protective effects.” page 7, line 19-21

“Second, lymphocyte deficiency was tissue protective in naïve mice without affecting cytokine levels, thus uncoupling early inflammation from tissue damage control in our model.” page 12, line 18-20

d) A key mechanism in LPS-tolerance is the upregulation of IRAK-M, which negatively regulates TLR signaling and thereby suppresses cytokine production. We have measured IRAK-M levels in livers of LPS pre-exposed mice and did not see any differences between naïve and tolerant animals (data not shown in the manuscript but can be provided upon request)

e) Another key mechanism of LPS-tolerance is the transcriptional suppression of inflammatory gene expression by the formation of NFkB p50 homodimers. We have tested mice deficient for the NFkB p50 subunit (NFkB^-/-^ mice) in our model. As expected, they were hyperresponsive to LPS, and half of the mice died from the LPS pretreatment. However, surviving animals still exhibited improved tissue damage control as compared to naïve controls during subsequent bacterial peritonitis (Figure 2K), suggesting that LPSinduced protection was independent of NFkB p50, again excluding classical LPStolerance as a mechanism for improved tissue damage control in tolerant mice.

We already mention this fact in our manuscript:

“… tissue damage control and LPS-induced disease tolerance during sepsis was induced independent of interferon-a/b receptor (IFNAR) signaling (Figure 2J and Figure 2—figure supplement 1I) and the anti-inflammatory NF-κB subunit p50 (NF-κB1) (Figure 2K and Figure 2—figure supplement 1J), which has been shown to mediate the suppression of cytokine production during endotoxin tolerance in vitro (Ziegler-Heitbrock 2001, Fan and Cook 2004).” Page 7, line 15-19

Taken together, while we observe suppressed early cytokine production, the cardinal feature of LPS-tolerance in our model, organ damage/protection was uncoupled from cytokine levels and furthermore was independent of key cellular (monocytes and macrophages) and molecular (NFkB p50) drivers of LPS-tolerance. At the same time, tissue damage control was improved without alterations in the pathogen load, which is why “disease tolerance” is the more appropriate term.

2) Please better explain the rationale of explanation leading from B-cell modulation to Cxcr4 and neutrophil activation

Thank you for this suggestion. We hypothesized a connection between B cells and neutrophils in the regulation of disease tolerance during *E. coli* sepsis because:

a) Deficiency of either B-cells or neutrophils abrogated sepsis-induced organ damage (Figure 2A, Figure 4A), suggesting a deleterious interaction of these two cell types in the regulation of tissue damage control.

To better explain this finding, we edited the Results section to now read:

“As the full deficiency of either B cells or neutrophils abrogated organ damage during *E. coli* sepsis and LPS-induced protection could be re-established by adoptive transfer of B cells into Rag2^-/-^ mice, we hypothesized an alliance between neutrophils and B cells in tissue damage control during sepsis.“ page 9, line 1-4

b) While splenectomy reduced tissue damage in general (i.e. in naïve mice), LPS pretreatment was still beneficial for splenectomized animals, suggesting that B cells in a compartment other than the spleen, were mediating the protective effects (Figure 2G). Moreover, the bone marrow B cell compartment was strongly affected by LPS treatment (prior induction of *E. coli* sepsis) (Figure 3).

To explain these findings the manuscript text reads:

“We then tested if splenectomy would replicate the protective effects of full B cell deficiency during sepsis and interestingly found that splenectomy was associated with reduced liver damage in naïve, sensitive mice, which is in line with other studies (Agarwal, Parant et al., 1972) but, in contrast to complete lymphocyte deficiency, not sufficient to abrogate LPS-induced tissue protection in tolerant animals (Figure 2G and Figure 2—figure supplement 1F). This suggested that mature splenic B cells contributed to tissue damage during severe infections, while other, not spleen derived, B cell compartments were instrumental in driving disease tolerance.” page 6, line 32-34, page 7, line 1-4

“Taken together, tissue damage control was associated with long-term changes in the B cell compartments in the spleen and bone marrow…" page 8, line 11-13

c) B cells and neutrophils are the most abundant cells in the murine bone marrow and are reciprocally (and functionally) regulated upon LPS challenge (Figure 4).

To address this fact, we state in the manuscript:

“In steady state, up to 70% of CD45+ bone marrow cells are composed of B cells and neutrophils, where both populations constitutively reside and mature by sharing the same niche” page 9, line 4-6

“We therefore first analyzed bone marrow B cell and neutrophil dynamics after LPS challenge, and discovered substantial stress-induced granulopoiesis, peaking around day four post LPS exposure, while B cells were regulated in a reciprocal fashion as they vanished by day four post LPS injection, to then increase and remain elevated two weeks post LPS treatment (Figure 4B and 3C) in tolerant, as compared to sensitive animals. At the same time total and relative neutrophil numbers in the bone marrow remained slightly reduced in tolerant wild type mice, but elevated in the absence of B cells (Figure 4C and Figure 4—figure supplement 1E)“ page 9, line 6-12

d) The idea that B cells and neutrophils are linked because they both utilize SDF1/Cxcr4 signaling was based on 3 key points:

– The SDF1/Cxcr4 axis is known to mediate B cell development and stem cell homeostasis as well as neutrophil retention in the bone marrow

– A study by Ueda et al., proposed competition of B lymphocytes and granulocytes for the same niche and enhanced sensitivity of B lymphocytes over neutrophils to drops in SDF1 levels (Ueda, Kondo et al., 2005). In a vaccination study, they convincingly showed that B cells and neutrophils regulate each other in a reciprocal fashion and that the decline in neutrophil numbers upon B cell expansion is directly dependent on B cells. They propose that this might be mediated via competition for SDF1.

– Our finding of elevated Cxcr4 expression in bone marrow neutrophils 2 weeks post LPS pretreatment and the absence of Cxcr4 upregulation in neutrophils of B cell deficient J_H_T mice, suggest an impact of B cells on neutrophil biology.

We already elaborate on these ideas extensively in our result section:

“Strikingly, bone marrow neutrophils of tolerant mice showed an enrichment of genes associated with cell migration, trafficking and chemotaxis (Figure 5C), such as genes involved in Cxcl12/Cxcr4 signaling including Cxcr4 itself (Figure 5—figure supplement 1E).

Considering the reported importance of Cxcr4 signaling in neutrophil trafficking between the bone marrow and periphery (Martin, Burdon et al., 2003, Eash, Greenbaum et al., 2010, Adrover, Del Fresno et al., 2019), we verified an upregulation of Cxcr4 on bone marrow derived neutrophils of tolerant mice compared to sensitive control mice on a transcriptional (Figure 5D) and protein level (Figure 5—figure supplement 1F). Importantly, this Cxcr4 induction depended on B cells, as Cxcr4 expression levels did not change in neutrophils isolated from LPS preexposed JHT mice (Figure 5D and Figure 5—figure supplement 1F). Based on these findings and the recent observation that Cxcr4 deficient neutrophils promote aging and neutrophilinduced vascular damage (Adrover, Del Fresno et al., 2019), we hypothesized that B cells impact the life cycle of neutrophils by influencing neutrophil Cxcr4 signaling, which in turn might promote tissue damage control during a subsequent sepsis.“ page 10, line 14-26

To further support this notion, we have now added a more detailed paragraph elaborating on this topic to our discussion:

“Cxcr4 interaction with its ligand Cxcl12 (stromal cell-derived factor 1, SDF1) has been shown to be critical for the retention of neutrophils in the bone marrow under steady state, their release to the periphery as well as their homing back to the bone marrow when they become senescent (Martin, Burdon et al., 2003, Eash, Greenbaum et al., 2010). Importantly, Cxcr4 signaling is essential, as Cxcr4 knockout mice die perinatally due to severe developmental defects ranging from virtually absent myelopoiesis and impaired B lymphopoiesis to abnormal brain development (Ma, Jones et al., 1998). A different sensitivity to changes in SDF1 concentrations as a potential mechanism of the reciprocal regulation of lymphopoiesis and granulopoiesis has been suggested earlier (Ueda, Kondo et al. 2005).“ page 13, line 33-34, page 14, line 1-7

3) The novelty of the paper needs to be clarified by putting the comments of Reviewer #2 into context.

In the revised manuscript, we have addressed most of Reviewer #2´s comments and stronger highlight the novelties of our study.

Reviewer #1 (Recommendations for the authors):– The authors discuss sepsis tolerance. However, the correct term is LPS tolerance. They are using the traditional LPS tolerance approach which is different from sepsis tolerance. Should they need to report on sepsis tolerance, all studied mice should have been subject to moderate bacteremia or to be survivors from CLP sepsis. This needs to be acknowledged in the Discussion and amended throughout the manuscript.

Thank you for this comment, we have addressed this issue extensively in the response to the editor (please see above) and have amended the manuscript to make this point clearer.

– The model the authors are using is based on survival benefit 24 hours following challenge with one small amount of LPS. Why then do the authors state that they have decided to study mice primed the last two weeks with LPS?

We believe that there is a misunderstanding as we do not mention any survival data in our manuscript but use tissue damage as a main readout. We have first tested the duration of LPS induced protection (ranging from 1d to 8 weeks) to then continue all studies using a 2 week window between LPS administration and induction of sepsis, to rule out any influence of an altered pathogen burden.

We have now adapted the text in hopes of better explaining the rationale behind our experimental setup (please also refer to our response to the editor):

“LPS pre-exposure has been shown to prevent hyperinflammation during fatal endotoxemia via the induction of LPS-tolerance (Lopez-Collazo and del Fresno 2013) and to improve the outcome of CLP sepsis by promoting bacterial clearance (Marquez-Velasco, Masso et al., 2007). To investigate mechanisms of tissue damage control and disease tolerance during bacterial sepsis we aimed for a setup with a stable pathogen load which would allow us to only interfere with sepsis-associated organ failure. We thus challenged mice intravenously (i.v.) with a subclinical dose of LPS 1 day, 2 weeks, 5 weeks or 8 weeks, respectively, prior to the induction of Gram-negative sepsis by intraperitoneal (i.p.) injection of the virulent *E. coli* strain O18:K1.“ Page 5, lines 3-9

“Thus, short-term (24h) LPS pre-exposure improved resistance to infection and consequently tissue integrity, while long-term (2-8 weeks) LPS pre-exposure enabled the maintenance of tissue integrity irrespective of a high bacterial load, which per definition resembles disease tolerance. To dissect the underlying mechanism of tissue damage control in disease tolerant mice, we thus performed all subsequent experiments by treating mice with either LPS or saline two weeks prior to bacterial infection, allowing us to compare tolerant with sensitive hosts. “ Page 5, lines 15-21

– I suggest that in parallel to sacrifice experiments showing the laboratory read-out of deleted elements of the immune system, some survival experiments are necessary particularly with the Rag2-/- mice.

The main phenotype we study is ‘organ protection’, i.e. an improved tissue damage control, and not survival differences. We decided to use organ damage parameters as a main readout (i) because it gives more mechanistic insight than survival (which can occur due to multiple reasons) and (ii) due to ethical reasons (survivals require more animals per group and cause much more suffering than humane endpoint experiments).

– LPS tolerance and modulation of the neutrophil phagocytosis requires decrease of the tissue bacterial load. Figure 1A does not suggest this.

As explained above, we did not look out for resistance mechanisms, i.e. reduced bacterial load, but instead wanted to focus on ‘disease tolerance’ irrespective of the bacterial load, which is why we on purpose chose a setup with a stable pathogen load using a virulent *E. coli* with a high growth rate, which is not further enhanced by the depletion of different immune cells. We also address this issue in the response to the editor.

– The authors need to better explain the rationale of explanation leading from B-cell modulation to Cxcr4 and neutrophil activation.

Thank you for this comment, we have addressed this issue in the response to the editor (please see above) and have amended the manuscript to explain the connection between Cxcr4 signaling, B cells and neutrophils in more detail.

Reviewer #2 (Recommendations for the authors):Although these data could certainly fine-draw the border of our knowledge on preconditioning/tolerance and sepsis-induced PMN changes in rodents, the study is largely recombinant and repetitive and the human/translational significance is also questionable.

In our revised manuscript we have highlighted the novelties of our study better and clarify that this is basic research performed in mice.

References used in this point by point reply:

Adrover, J. M., C. Del Fresno, G. Crainiciuc, M. I. Cuartero, M. Casanova-Acebes, L. A. Weiss, H. Huerga-Encabo, C. Silvestre-Roig, J. Rossaint, I. Cossio, A. V. Lechuga-Vieco, J. Garcia-Prieto, M. Gomez-Parrizas, J. A. Quintana, I. Ballesteros, S. Martin-Salamanca, A. Aroca-Crevillen, S. Z. Chong, M. Evrard, K. Balabanian, J. Lopez, K. Bidzhekov, F. Bachelerie, F. Abad-Santos, C. Munoz-Calleja, A. Zarbock, O. Soehnlein, C. Weber, L. G. Ng, C. Lopez-Rodriguez, D. Sancho, M. A. Moro, B. Ibanez and A. Hidalgo (2019). "A Neutrophil Timer Coordinates Immune Defense and Vascular Protection." Immunity 50(2): 390-402 e310.

Agarwal, M. K., M. Parant and F. Parant (1972). "Role of spleen in endotoxin poisoning and reticuloendothelial function." Br J Exp Pathol 53(5): 485-491.

Beutler, B. (2000). "Tlr4: central component of the sole mammalian LPS sensor." Curr Opin Immunol 12(1): 20-26.

Bianconi, V., A. Sahebkar, S. L. Atkin and M. Pirro (2018). "The regulation and importance of monocyte chemoattractant protein-1." Curr Opin Hematol 25(1): 44-51.

de Vos, A. F., J. M. Pater, P. S. van den Pangaart, M. D. de Kruif, C. van 't Veer and T. van der Poll (2009). "in vivo lipopolysaccharide exposure of human blood leukocytes induces crosstolerance to multiple TLR ligands." J Immunol 183(1): 533-542.

Divangahi, M., P. Aaby, S. A. Khader, L. B. Barreiro, S. Bekkering, T. Chavakis, R. van Crevel, N. Curtis, A. R. DiNardo, J. Dominguez-Andres, R. Duivenvoorden, S. Fanucchi, Z. Fayad, E. Fuchs, M. Hamon, K. L. Jeffrey, N. Khan, L. A. B. Joosten, E. Kaufmann, E. Latz, G. Matarese, J. W. M. van der Meer, M. Mhlanga, S. Moorlag, W. J. M. Mulder, S. Naik, B. Novakovic, L. O'Neill, J. Ochando, K. Ozato, N. P. Riksen, R. Sauerwein, E. R. Sherwood, A. Schlitzer, J. L. Schultze, M.

H. Sieweke, C. S. Benn, H. Stunnenberg, J. Sun, F. L. van de Veerdonk, S. Weis, D. L. Williams, R. Xavier and M. G. Netea (2021). "Trained immunity, tolerance, priming and differentiation: distinct immunological processes." Nat Immunol 22(1): 2-6.

Eash, K. J., A. M. Greenbaum, P. K. Gopalan and D. C. Link (2010). "CXCR2 and CXCR4 antagonistically regulate neutrophil trafficking from murine bone marrow." J Clin Invest 120(7): 2423-2431.

Fan, H. and J. A. Cook (2004). "Molecular mechanisms of endotoxin tolerance." J Endotoxin Res 10(2): 71-84.

Freudenberg, M. A. and C. Galanos (1988). "Induction of tolerance to lipopolysaccharide (LPS)D-galactosamine lethality by pretreatment with LPS is mediated by macrophages." Infect Immun 56(5): 1352-1357.

Gawish, R., R. Martins, B. Bohm, T. Wimberger, O. Sharif, K. Lakovits, M. Schmidt and S. Knapp (2015). "Triggering receptor expressed on myeloid cells-2 fine-tunes inflammatory responses in murine Gram-negative sepsis." FASEB J 29(4): 1247-1257.

Hagar, J. A., D. A. Powell, Y. Aachoui, R. K. Ernst and E. A. Miao (2013). "Cytoplasmic LPS activates caspase-11: implications in TLR4-independent endotoxic shock." Science 341(6151): 1250-1253.

Hoshino, K., O. Takeuchi, T. Kawai, H. Sanjo, T. Ogawa, Y. Takeda, K. Takeda and S. Akira (1999). "Cutting edge: Toll-like receptor 4 (TLR4)-deficient mice are hyporesponsive to lipopolysaccharide: evidence for TLR4 as the Lps gene product." J Immunol 162(7): 3749-3752. Hotchkiss, R. S., G. Monneret and D. Payen (2013). "Sepsis-induced immunosuppression: from cellular dysfunctions to immunotherapy." Nat Rev Immunol 13(12): 862-874.

Hotchkiss, R. S., P. E. Swanson, J. P. Cobb, A. Jacobson, T. G. Buchman and I. E. Karl (1997). "Apoptosis in lymphoid and parenchymal cells during sepsis: findings in normal and T- and Bcell-deficient mice." Crit Care Med 25(8): 1298-1307.

Hotchkiss, R. S., P. E. Swanson, C. M. Knudson, K. C. Chang, J. P. Cobb, D. F. Osborne, K. M. Zollner, T. G. Buchman, S. J. Korsmeyer and I. E. Karl (1999). "Overexpression of BCl^-^2 in transgenic mice decreases apoptosis and improves survival in sepsis." J Immunol 162(7): 41484156.

Karanfilian, R. G., C. R. Spillert, G. W. Machiedo, B. F. Rush, Jr. and E. J. Lazaro (1983). "Effect of age and splenectomy in murine endotoxemia." Adv Shock Res 9: 125-132.

Kayagaki, N., M. T. Wong, I. B. Stowe, S. R. Ramani, L. C. Gonzalez, S. Akashi-Takamura, K. Miyake, J. Zhang, W. P. Lee, A. Muszynski, L. S. Forsberg, R. W. Carlson and V. M. Dixit (2013). "Noncanonical inflammasome activation by intracellular LPS independent of TLR4." Science 341(6151): 1246-1249.

Kelly-Scumpia, K. M., P. O. Scumpia, J. S. Weinstein, M. J. Delano, A. G. Cuenca, D. C. Nacionales, J. L. Wynn, P. Y. Lee, Y. Kumagai, P. A. Efron, S. Akira, C. Wasserfall, M. A. Atkinson and L. L. Moldawer (2011). "B cells enhance early innate immune responses during bacterial sepsis." J Exp Med 208(8): 1673-1682.

Knapp, S., A. F. de Vos, S. Florquin, D. T. Golenbock and T. van der Poll (2003). "Lipopolysaccharide binding protein is an essential component of the innate immune response to *Escherichia coli* peritonitis in mice." Infect Immun 71(12): 6747-6753.

Knapp, S., U. Matt, N. Leitinger and T. van der Poll (2007). "Oxidized phospholipids inhibit phagocytosis and impair outcome in gram-negative sepsis in vivo." J Immunol 178(2): 9931001.

Liu, Q., Z. Li, J. L. Gao, W. Wan, S. Ganesan, D. H. McDermott and P. M. Murphy (2015). "CXCR4 antagonist AMD3100 redistributes leukocytes from primary immune organs to secondary immune organs, lung, and blood in mice." Eur J Immunol 45(6): 1855-1867.

Lopez-Collazo, E. and C. del Fresno (2013). "Pathophysiology of endotoxin tolerance:

mechanisms and clinical consequences." Crit Care 17(6): 242.

Ma, Q., D. Jones, P. R. Borghesani, R. A. Segal, T. Nagasawa, T. Kishimoto, R. T. Bronson and T. A. Springer (1998). "Impaired B-lymphopoiesis, myelopoiesis, and derailed cerebellar neuron migration in CXCR4- and SDF-1-deficient mice." Proc Natl Acad Sci U S A 95(16): 9448-9453. Marquez-Velasco, R., F. Masso, R. Hernandez-Pando, L. F. Montano, R. Springall, L. M. Amezcua-Guerra and R. Bojalil (2007). "LPS pretreatment by the oral route protects against sepsis induced by cecal ligation and puncture. Regulation of proinflammatory response and IgM anti-LPS antibody production as associated mechanisms." Inflamm Res 56(9): 385-390.

Martin, C., P. C. Burdon, G. Bridger, J. C. Gutierrez-Ramos, T. J. Williams and S. M. Rankin (2003). "Chemokines acting via CXCR2 and CXCR4 control the release of neutrophils from the bone marrow and their return following senescence." Immunity 19(4): 583-593.

Martins, R., A. R. Carlos, F. Braza, J. A. Thompson, P. Bastos-Amador, S. Ramos and M. P. Soares (2019). "Disease Tolerance as an Inherent Component of Immunity." Annu Rev Immunol 37: 405-437.

Martins, R., A. R. Carlos, F. Braza, J. A. Thompson, P. Bastos-Amador, S. Ramos and M. P. Soares (2019). "Disease Tolerance as an Inherent Component of Immunity." Annu Rev Immunol. McCarville, J. L. and J. S. Ayres (2018). "Disease tolerance: concept and mechanisms." Curr Opin Immunol 50: 88-93.

Medzhitov, R., D. S. Schneider and M. P. Soares (2012). "Disease tolerance as a defense strategy." Science 335(6071): 936-941.

Ngamsri, K. C., C. Jans, R. A. Putri, K. Schindler, J. Gamper-Tsigaras, C. Eggstein, D. Kohler and F. M. Konrad (2020). "Inhibition of CXCR4 and CXCR7 Is Protective in Acute Peritoneal Inflammation." Front Immunol 11: 407.

Rajarathnam, K., M. Schnoor, R. M. Richardson and S. Rajagopal (2019). "How do chemokines navigate neutrophils to the target site: Dissecting the structural mechanisms and signaling pathways." Cell Signal 54: 69-80.

Sano, T., K. Izuishi, M. A. Hossain, K. Kakinoki, K. Okano, T. Masaki and Y. Suzuki (2010). "Protective effect of lipopolysaccharide preconditioning in hepatic ischaemia reperfusion injury." HPB (Oxford) 12(8): 538-545.

Schafer, J. F. (1971). "Tolerance to Plant Disease." Annual Review of Phytopathology 9(1): 235252.

Schneider, D. S. (2021). "Immunology's intolerance of disease tolerance." Nat Rev Immunol 21(10): 624-625.

Ueda, Y., M. Kondo and G. Kelsoe (2005). "Inflammation and the reciprocal production of granulocytes and lymphocytes in bone marrow." J Exp Med 201(11): 1771-1780.

Yang, M., G. Busche, A. Ganser and Z. Li (2013). "Morphology and quantitative composition of hematopoietic cells in murine bone marrow and spleen of healthy subjects." Ann Hematol 92(5): 587-594.

Zahalka, S., P. Starkl, M. L. Watzenboeck, A. Farhat, M. Radhouani, F. Deckert, A. Hladik, K. Lakovits, F. Oberndorfer, C. Lassnig, B. Strobl, K. Klavins, M. Matsushita, D. E. Sanin, K. M. Grzes, E. J. Pearce, A.-D. Gorki and S. Knapp (2022). "Trained immunity of alveolar macrophages requires metabolic rewiring and type 1 interferon signaling." Mucosal Immunology.

Zeisberger, E. and J. Roth (1998). "Tolerance to pyrogens." Ann N Y Acad Sci 856: 116-131. Ziegler-Heitbrock, L. (2001). "The p50-homodimer mechanism in tolerance to LPS." J Endotoxin Res 7(3): 219-222.